# Diffusion-LM Improves Controllable Text Generation

**Xiang Lisa Li**
Stanford University
xlisali@stanford.edu

**John Thickstun**
Stanford University
jthickst@stanford.edu

**Ishaan Gulrajani**
Stanford Univeristy
igul@stanford.edu

**Percy Liang**
Stanford Univeristy
pliang@cs.stanford.edu

**Tatsunori B. Hashimoto**
Stanford Univeristy
thashim@stanford.edu

## Abstract

Controlling the behavior of language models (LMs) without re-training is a major open problem in natural language generation. While recent works have demonstrated successes on controlling simple sentence attributes (e.g., sentiment), there has been little progress on complex, fine-grained controls (e.g., syntactic structure). To address this challenge, we develop a new *non-autoregressive* language model based on *continuous* diffusions that we call Diffusion-LM. Building upon the recent successes of diffusion models in continuous domains, Diffusion-LM iteratively denoises a sequence of Gaussian vectors into word vectors, yielding a sequence of intermediate latent variables. The continuous, hierarchical nature of these intermediate variables enables a simple gradient-based algorithm to perform complex, controllable generation tasks. We demonstrate successful control of Diffusion-LM for six challenging fine-grained control tasks, significantly outperforming prior work.[1]

## 1 Introduction

Large autoregressive language models (LMs) are capable of generating high quality text [39, 3, 5, 56], but in order to reliably deploy these LMs in real world applications, the text generation process needs to be *controllable*: we need to generate text that satisfies desired requirements (e.g. topic, syntactic structure). A natural approach for controlling a LM would be to fine-tune the LM using supervised data of the form (control, text) [18]. However, updating the LM parameters for each control task can be expensive and does not allow for compositions of multiple controls (e.g. generate text that is both positive sentiment *and* non-toxic). This motivates light-weight and modular plug-and-play approaches [6] that keep the LM frozen and steer the generation process using an external classifier that measures how well the generated text satisfies the control. But steering a frozen autoregressive LM has been shown to be difficult, and existing successes have been limited to simple, attribute-level controls (e.g., sentiment or topic) [6, 25, 55].

In order to tackle more complex controls, we propose *Diffusion-LM*, a new language model based on *continuous* diffusions. Diffusion-LM starts with a sequence of Gaussian noise vectors and incrementally denoises them into vectors corresponding to words, as shown in Figure 1. These gradual denoising steps produce a hierarchy of continuous latent representations. We find that this hierarchical and continuous latent variable enables simple, gradient-based methods to perform complex control tasks such as constraining the parse tree of a generated sequence.

Continuous diffusion models have been extremely successful in vision and audio domains [13, 24, 41, 8, 4], but they have not been applied to text because of the inherently discrete nature of text

---

[1]Code is available at https://github.com/XiangLi1999/Diffusion-LM.git

36th Conference on Neural Information Processing Systems (NeurIPS 2022).

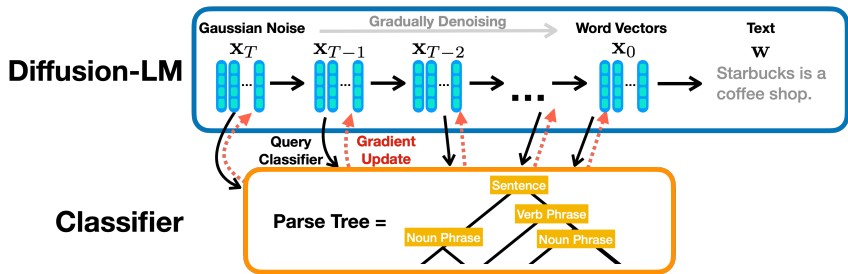

Figure 1: Diffusion-LM iteratively denoises a sequence of Gaussian vectors into word vectors, yielding a intermediate latent variables of decreasing noise level $\mathbf{x}_T \cdots \mathbf{x}_0$. For controllable generation, we iteratively perform gradient updates on these continuous latents to optimize for fluency (parametrized by Diffusion-LM) and satisfy control requirements (parametrized by a classifier).

(§3). Adapting this class of models to text requires several modifications to standard diffusions: we add an embedding step and a rounding step to the standard diffusion process, design a training objective to learn the embedding, and propose techniques to improve rounding (§4). We control Diffusion-LM using a gradient-based method, as shown in Figure 1. This method enables us to steer the text generation process towards outputs that satisfy target structural and semantic controls. It iteratively performs gradient updates on the continuous latent variables of Diffusion-LM to balance fluency and control satisfaction (§5.1).

To demonstrate control of Diffusion-LM, we consider six control targets ranging from fine-grained attributes (e.g., semantic content) to complex structures (e.g., parse trees). Our method almost doubles the success rate of previous plug-and-play methods and matches or outperforms the fine-tuning oracle on all these classifier-guided control tasks (§7.1). In addition to these individual control tasks, we show that we can successfully compose multiple classifier-guided controls to generate sentences with both desired semantic content and syntactic structure (§7.2). Finally, we consider span-anchored controls, such as length control and infilling. Diffusion-LM allows us to perform these control tasks *without* a classifier, and our Diffusion-LM significantly outperforms prior plug-and-play methods and is on-par with an autoregressive LM trained from scratch for the infilling task (§7.3).

## 2   Related Work

**Diffusion Models for Text.**   Diffusion models [47] have demonstrated great success in continuous data domains [13, 33, 24, 31], producing images and audio that have state-of-the-art sample quality. To handle discrete data, past works have studied text diffusion models on *discrete* state spaces, which defines a corruption process on discrete data (e.g., each token has some probability to be corrupted to an absorbing or random token) [1, 15, 16]. In this paper, we focus on *continuous* diffusion models for text and to the best of our knowledge, our work is the first to explore this setting. In contrast to discrete diffusion LMs, our continuous diffusion LMs induce continuous latent representations, which enables efficient gradient-based methods for controllable generation.

**Autoregressive and Non-autoregressive LMs.**   Most large pre-trained LMs are left-to-right autoregressive (e.g., GPT-3 [3], PaLM [5]). The fixed generation order limits the models' flexibility in many controllable generation settings, especially those that impose controls globally on both left and right contexts. One example is infilling, which imposes lexical control on the right contexts; another example is syntactic structure control, which controls global properties involving both left and right contexts. Since autoregressive LMs cannot directly condition on right contexts, prior works have developed specialized training and decoding techniques for these tasks [46, 9, 36]. For example, Qin et al. [37] proposed a decoding method that relaxes the discrete LM outputs to continuous variables and backpropagates gradient information from the right context. Diffusion-LM can condition on arbitrary classifiers that look at complex, global properties of the sentence. There are other non-autoregressive LMs that have been developed for machine translation and speech-to-text tasks [12, 43]. However these methods are specialized for speech and translation settings, where the entropy over valid outputs is low, and whether they work for language modeling remains an open problem. We leave detailed discussions to Appendix H.

**Plug-and-Play Controllable Generation.**   Plug-and-play controllable generation aims to keep the LM frozen and steer its output using potential functions (e.g., classifiers). Given a probabilistic

potential function that measures how well the generated text satisfies the desired control, the generated text should be optimized for both control satisfaction (measured by the potential function) and fluency (measured by LM probabilities) . There are several plug-and-play approaches based on autoregressive LMs: FUDGE [55] reweights the LM prediction at each token with an estimate of control satisfaction for the partial sequence; GeDi [25] and DExperts [28] reweight the LM prediction at each token with a smaller LM finetuned/trained for the control task.

The closest work to ours is PPLM [6], which runs gradient ascent on an autoregressive LM's hidden activations to steer the next token to satisfy the control and maintain fluency. Because PPLM is based on autoregressive LMs, it can only generate left-to-right. This prevents PPLM from repairing and recovering errors made in previous generation steps. Despite their success on attribute (e.g., topic) controls, we will show these plug-and-play methods for autoregressive LMs fail on more complex control tasks such as controlling syntactic structure and semantic content in §7.1. We demonstrate that Diffusion-LM is capable of plug-and-play controllable generation by applying classifier-guided gradient updates to the continuous sequence of latent variables induced by the Diffusion-LM.

## 3 Problem Statement and Background

We first define controllable generation (§3.1) and then review continuous diffusion models (§3.3).

### 3.1 Generative Models and Controllable Generation for Text

Text generation is the task of sampling $\mathbf{w}$ from a trained language model $p_{\text{lm}}(\mathbf{w})$, where $\mathbf{w} = [w_1 \cdots w_n]$ is a sequence of discrete words and $p_{\text{lm}}(\mathbf{w})$ is a probability distribution over sequences of words. Controllable text generation is the task of sampling $\mathbf{w}$ from a conditional distribution $p(\mathbf{w} \mid \mathbf{c})$, where $\mathbf{c}$ denotes a *control* variable. For syntactic control, $\mathbf{c}$ can be a target syntax tree (Figure 1), while for sentiment control, $\mathbf{c}$ could be a desired sentiment label. The goal of controllable generation is to generate $\mathbf{w}$ that satisfies the control target $\mathbf{c}$.

Consider the plug-and-play controllable generation setting: we are given a language model $p_{\text{lm}}(\mathbf{w})$ trained from a large amount of unlabeled text data, and for each control task, we are given a classifier $p(\mathbf{c} \mid \mathbf{w})$ trained from smaller amount of labeled text data (e.g., for syntactic control, the classifier is a probabilistic parser). The goal is to utilize these two models to approximately sample from the posterior $p(\mathbf{w} \mid \mathbf{c})$ via Bayes rule $p(\mathbf{w} \mid \mathbf{c}) \propto p_{\text{lm}}(\mathbf{w}) \cdot p(\mathbf{c} \mid \mathbf{w})$. Here, $p_{\text{lm}}(\mathbf{w})$ encourages $\mathbf{w}$ to be fluent, and the $p(\mathbf{c} \mid \mathbf{w})$ encourages $\mathbf{w}$ to fulfill the control.

### 3.2 Autoregressive Language Models

The canonical approach to language modeling factors $p_{\text{lm}}$ in an autoregressive left-to-right mannar, $p_{\text{lm}}(\mathbf{w}) = p_{\text{lm}}(w_1) \prod_{i=2}^{n} p_{\text{lm}}(x_i \mid x_{<i})$. In this case, text generation is reduced to the task of repeatedly predicting the next token conditioned on the partial sequence generated so far. The next token prediction $p_{\text{lm}}(x_i \mid x_{<i})$ is often parametrized by Transformer architecture [52].

### 3.3 Diffusion Models for Continuous Domains

A diffusion model [13, 33] is a latent variable model that models the data $\mathbf{x}_0 \in \mathbb{R}^d$ as a Markov chain $\mathbf{x}_T \ldots \mathbf{x}_0$ with each variable in $\mathbb{R}^d$, and $\mathbf{x}_T$ is a Gaussian. The diffusion model incrementally denoises the sequence of latent variables $\mathbf{x}_{T:1}$ to approximate samples from the target data distribution (Figure 2). The initial state $p_\theta(\mathbf{x}_T) \approx \mathcal{N}(0, \mathbf{I})$, and each denoising transition $\mathbf{x}_t \to \mathbf{x}_{t-1}$ is parametrized by the model $p_\theta(\mathbf{x}_{t-1} \mid \mathbf{x}_t) = \mathcal{N}(\mathbf{x}_{t-1}; \mu_\theta(\mathbf{x}_t, t), \Sigma_\theta(\mathbf{x}_t, t))$. For example, $\mu_\theta$ and $\Sigma_\theta$ may be computed by a U-Net or a Tranformer.

To train the diffusion model, we define a forward process that constructs the intermediate latent variables $\mathbf{x}_{1:T}$. The forward process incrementally adds Gaussian noise to data $\mathbf{x}_0$ until, at diffusion step $T$, samples $\mathbf{x}_T$ are approximately Gaussian. Each transition $\mathbf{x}_{t-1} \to \mathbf{x}_t$ is parametrized by $q(\mathbf{x}_t \mid \mathbf{x}_{t-1}) = \mathcal{N}(\mathbf{x}_t; \sqrt{1 - \beta_t}\mathbf{x}_{t-1}, \beta_t \mathbf{I})$, where the hyperparameter $\beta_t$ is the amount of noise added at diffusion step $t$. This parametrization of the forward process $q$ contains no trainable parameters and allows us to define a training objective that involves generating noisy data according to a pre-defined forward process $q$ and training a model to reverse the process and reconstruct the data.

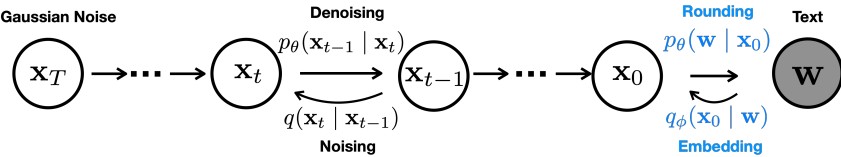

Figure 2: A graphical model representing the forward and reverse diffusion processes. In addition to the original diffusion models [13], we add a Markov transition between $\mathbf{x}_0$ and $\mathbf{w}$, and propose the embedding §4.1 and rounding §4.2 techniques.

The diffusion model is trained to maximize the marginal likelihood of the data $\mathbb{E}_{\mathbf{x}_0 \sim p_{\text{data}}}[\log p_\theta(\mathbf{x}_0)]$, and the canonical objective is the variational lower bound of $\log p_\theta(\mathbf{x}_0)$ [47],

$$\mathcal{L}_{\text{vlb}}(\mathbf{x}_0) = \mathop{\mathbb{E}}_{q(\mathbf{x}_{1:T}|\mathbf{x}_0)} \left[ \log \frac{q(\mathbf{x}_T|\mathbf{x}_0)}{p_\theta(\mathbf{x}_T)} + \sum_{t=2}^{T} \log \frac{q(\mathbf{x}_{t-1}|\mathbf{x}_0, \mathbf{x}_t)}{p_\theta(\mathbf{x}_{t-1}|\mathbf{x}_t)} - \log p_\theta(\mathbf{x}_0|\mathbf{x}_1) \right]. \tag{1}$$

However, this objective can be unstable and require many optimization tricks to stabilize [33]. To circumvent this issue, Ho et al. [13] devised a simple surrogate objective that expands and reweights each KL-divergence term in $\mathcal{L}_{\text{vlb}}$ to obtain a mean-squared error loss (derivation in Appendix J) which we will refer to as

$$\mathcal{L}_{\text{simple}}(\mathbf{x}_0) = \sum_{t=1}^{T} \mathop{\mathbb{E}}_{q(\mathbf{x}_t|\mathbf{x}_0)} ||\mu_\theta(\mathbf{x}_t, t) - \hat{\mu}(\mathbf{x}_t, \mathbf{x}_0)||^2,$$

where $\hat{\mu}(\mathbf{x}_t, \mathbf{x}_0)$ is the mean of the posterior $q(\mathbf{x}_{t-1}|\mathbf{x}_0, \mathbf{x}_t)$ which is a closed from Gaussian, and $\mu_\theta(\mathbf{x}_t, t)$ is the predicted mean of $p_\theta(\mathbf{x}_{t-1} \mid \mathbf{x}_t)$ computed by a neural network. While $\mathcal{L}_{\text{simple}}$ is no longer a valid lower bound, prior work has found that it empirically made training more stable and improved sample quality[2]. We will make use of similar simplifications in Diffusion-LM to stabilize training and improve sample quality (§4.1).

## 4 Diffusion-LM: Continuous Diffusion Language Modeling

Constructing Diffusion-LM requires several modifications to the standard diffusion model. First, we must define an embedding function that maps discrete text into a continuous space. To address this, we propose an end-to-end training objective for learning embeddings (§4.1). Second, we require a rounding method to map vectors in embedding space back to words. To address this, we propose training and decoding time methods to facilitate rounding (§4.2).

### 4.1 End-to-end Training

To apply a continuous diffusion model to discrete text, we define an embedding function $\text{EMB}(w_i)$ that maps each word to a vector in $\mathbb{R}^d$. We define the embedding of a sequence $\mathbf{w}$ of length $n$ to be: $\text{EMB}(\mathbf{w}) = [\text{EMB}(w_1), \ldots, \text{EMB}(w_n)] \in \mathbb{R}^{nd}$.

We propose a modification of the diffusion model training objective (Equation 1) that jointly learns the diffusion model's parameters and word embeddings. In preliminary experiments, we explored random Gaussian embeddings, as well as pre-trained word embeddings [35, 39]. We found that these fixed embeddings are suboptimal for Diffusion-LM compared to end-to-end training[3].

As shown in Figure 2, our approach adds a Markov transition from discrete words $\mathbf{w}$ to $\mathbf{x}_0$ in the forward process, parametrized by $q_\phi(\mathbf{x}_0|\mathbf{w}) = \mathcal{N}(\text{EMB}(\mathbf{w}), \sigma_0 I)$. In the reverse process, we add a trainable rounding step, parametrized by $p_\theta(\mathbf{w} \mid \mathbf{x}_0) = \prod_{i=1}^{n} p_\theta(w_i \mid x_i)$, where $p_\theta(w_i \mid x_i)$ is a softmax distribution. The training objectives introduced in §3 now becomes

$$\begin{aligned} \mathcal{L}_{\text{vlb}}^{\text{e2e}}(\mathbf{w}) &= \mathop{\mathbb{E}}_{q_\phi(\mathbf{x}_0|\mathbf{w})} \left[ \mathcal{L}_{\text{vlb}}(\mathbf{x}_0) + \log q_\phi(\mathbf{x}_0|\mathbf{w}) - \log p_\theta(\mathbf{w}|\mathbf{x}_0)] \right], \\ \mathcal{L}_{\text{simple}}^{\text{e2e}}(\mathbf{w}) &= \mathop{\mathbb{E}}_{q_\phi(\mathbf{x}_{0:T}|\mathbf{w})} \left[ \mathcal{L}_{\text{simple}}(\mathbf{x}_0) + ||\text{EMB}(\mathbf{w}) - \mu_\theta(\mathbf{x}_1, 1)||^2 - \log p_\theta(\mathbf{w}|\mathbf{x}_0) \right]. \end{aligned} \tag{2}$$

---

[2]Our definition of $\mathcal{L}_{\text{simple}}$ here uses a different parametrization from Ho et al. [13]. We define our squared loss in terms of $\mu_\theta(\mathbf{x}_t, t)$ while they express it in terms of $\epsilon_\theta(\mathbf{x}_t, t)$.

[3]While trainable embeddings perform best on control and generation tasks, we found that fixed embeddings onto the vocabulary simplex were helpful when optimizing for held-out perplexity. We leave discussion of this approach and perplexity results to Appendix K as the focus of this work is generation quality and not perplexity.

We derive $\mathcal{L}^{\text{e2e}}_{\text{simple}}(\mathbf{w})$ from $\mathcal{L}^{\text{e2e}}_{\text{vlb}}(\mathbf{w})$ following the simplification in §3.3 and our derivation details are in Appendix J. Since we are training the embedding function, $q_\phi$ now contains trainable parameters and we use the reparametrization trick [42, 20] to backpropagate through this sampling step. Empirically, we find the learned embeddings cluster meaningfully: words with the same part-of-speech tags (syntactic role) tend to be clustered, as shown in Figure 3.

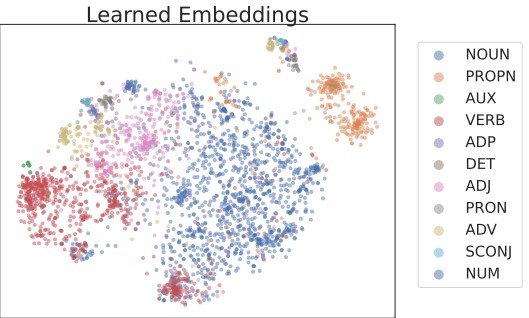

Figure 3: A t-SNE [51] plot of the learned word embeddings. Each word is colored by its POS.

## 4.2 Reducing Rounding Errors

The learned embeddings define a mapping from discrete text to the continuous $\mathbf{x}_0$. We now describe the inverse process of rounding a predicted $\mathbf{x}_0$ back to discrete text. Rounding is achieved by choosing the most probable word for each position, according to $\text{argmax}\, p_\theta(\mathbf{w} \mid \mathbf{x}_0) = \prod_{i=1}^n p_\theta(w_i \mid x_i)$. Ideally, this argmax-rounding would be sufficient to map back to discrete text, as the denoising steps should ensure that $\mathbf{x}_0$ lies exactly on the embedding of some word. However, empirically, the model fails to generate $\mathbf{x}_0$ that commits to a single word.

One explanation for this phenomenon is that the $\mathcal{L}_{\text{simple}}(\mathbf{x}_0)$ term in our objective 2 puts insufficient emphasis on modeling the structure of $\mathbf{x}_0$. Recall that we defined $\mathcal{L}_{\text{simple}}(\mathbf{x}_0) = \sum_{t=1}^T \mathbb{E}_{\mathbf{x}_t} ||\mu_\theta(\mathbf{x}_t, t) - \hat{\mu}(\mathbf{x}_t, \mathbf{x}_0)||^2$, where our model $\mu_\theta(\mathbf{x}_t, t)$ directly predicts the mean of $p_\theta(\mathbf{x}_{t-1} \mid \mathbf{x}_t)$ for each denoising step $t$. In this objective, the constraint that $\mathbf{x}_0$ has to commit to a single word embedding will only appear in the terms with $t$ near 0, and we found that this parametrization required careful tuning to force the objective to emphasize those terms (see Appendix M).

Our approach re-parametrizes $\mathcal{L}_{\text{simple}}$ to force Diffusion-LM to explicitly model $\mathbf{x}_0$ in *every* term of the objective. Specifically, we derive an analogue to $\mathcal{L}_{\text{simple}}$ which is parametrized via $\mathbf{x}_0$, $\mathcal{L}^{\text{e2e}}_{\mathbf{x}_0\text{-simple}}(\mathbf{x}_0) = \sum_{t=1}^T \mathbb{E}_{\mathbf{x}_t} ||f_\theta(\mathbf{x}_t, t) - \mathbf{x}_0||^2$, where our model $f_\theta(\mathbf{x}_t, t)$ predicts $\mathbf{x}_0$ directly [4]. This forces the neural network to predict $\mathbf{x}_0$ in every term and we found that models trained with this objective quickly learn that $\mathbf{x}_0$ should precisely centered at a word embedding.

We described how re-parametrization can be helpful for model training, but we also found that the same intuition could be used at decoding time in a technique that we call the *clamping* trick. In the standard generation approach for a $\mathbf{x}_0$-parametrized model, the model denoises $\mathbf{x}_t$ to $\mathbf{x}_{t-1}$ by first computing an estimate of $\mathbf{x}_0$ via $f_\theta(\mathbf{x}_t, t)$ and then sampling $\mathbf{x}_{t-1}$ conditioned on this estimate: $\mathbf{x}_{t-1} = \sqrt{\bar{\alpha}} f_\theta(\mathbf{x}_t, t) + \sqrt{1 - \bar{\alpha}} \epsilon$, where $\bar{\alpha}_t = \prod_{s=0}^t (1 - \beta_s)$ and $\epsilon \sim \mathcal{N}(0, I)$ [5]. In the clamping trick, the model additionally maps the predicted vector $f_\theta(\mathbf{x}_t, t)$ to its nearest word embedding sequence. Now, the sampling step becomes $\mathbf{x}_{t-1} = \sqrt{\bar{\alpha}} \cdot \text{Clamp}(f_\theta(\mathbf{x}_t, t)) + \sqrt{1 - \bar{\alpha}} \epsilon$. The clamping trick forces the predicted vector to commit to a word for intermediate diffusion steps, making the vector predictions more precise and reducing rounding errors.[6]

## 5 Decoding and Controllable Generation with Diffusion-LM

Having described the Diffusion-LM, we now consider the problem of controllable text generation (§5.1) and decoding (§5.2).

---

[4]Predicting $\mathbf{x}_0$ and $\mathbf{x}_{t-1}$ is equivalent up to scaling constants as the distribution of $\mathbf{x}_{t-1}$ can be obtained in closed form via the forward process $\mathbf{x}_{t-1} = \sqrt{\bar{\alpha}} \mathbf{x}_0 + \sqrt{1 - \bar{\alpha}} \epsilon$, see Appendix J for further details.

[5]This follows from the marginal distribution $q(\mathbf{x}_t \mid \mathbf{x}_0)$, which is a closed form Gaussian since all the Markov transitions are Gaussian.

[6]Intuitively, applying the clamping trick to early diffusion steps with $t$ near $T$ may be sub-optimal, because the model hasn't figured out what words to commit to. Empirically, applying clamping trick for all diffusion steps doesn't hurt the performance much. But to follow this intuition, one could also set the starting step of the clamping trick as a hyperparameter.

## 5.1 Controllable Text Generation

We now describe a procedure that enables plug-and-play control on Diffusion-LM. Our approach to control is inspired by the Bayesian formulation in §3.1, but instead of performing control directly on the discrete text, we perform control on the sequence of continuous latent variables $\mathbf{x}_{0:T}$ defined by Diffusion-LM, and apply the rounding step to convert these latents into text.

Controlling $\mathbf{x}_{0:T}$ is equivalent to decoding from the posterior $p(\mathbf{x}_{0:T}|\mathbf{c}) = \prod_{t=1}^{T} p(\mathbf{x}_{t-1} \mid \mathbf{x}_t, \mathbf{c})$, and we decompose this joint inference problem to a sequence of control problems at each diffusion step: $p(\mathbf{x}_{t-1} \mid \mathbf{x}_t, \mathbf{c}) \propto p(\mathbf{x}_{t-1} \mid \mathbf{x}_t) \cdot p(\mathbf{c} \mid \mathbf{x}_{t-1}, \mathbf{x}_t)$. We further simplify $p(\mathbf{c} \mid \mathbf{x}_{t-1}, \mathbf{x}_t) = p(\mathbf{c} \mid \mathbf{x}_{t-1})$ via conditional independence assumptions from prior work on controlling diffusions [49]. Consequently, for the $t$-th step, we run gradient update on $\mathbf{x}_{t-1}$:

$$\nabla_{\mathbf{x}_{t-1}} \log p(\mathbf{x}_{t-1} \mid \mathbf{x}_t, \mathbf{c}) = \nabla_{\mathbf{x}_{t-1}} \log p(\mathbf{x}_{t-1} \mid \mathbf{x}_t) + \nabla_{\mathbf{x}_{t-1}} \log p(\mathbf{c} \mid \mathbf{x}_{t-1}),$$

where both $\log p(\mathbf{x}_{t-1} \mid \mathbf{x}_t)$ and $\log p(\mathbf{c} \mid \mathbf{x}_{t-1})$ are differentiable: the first term is parametrized by Diffusion-LM, and the second term is parametrized by a neural network classifier.

Similar to work in the image setting [8, 49], we train the classifier on the diffusion latent variables and run gradient updates on the latent space $\mathbf{x}_{t-1}$ to steer it towards fulfilling the control. These image diffusion works take one gradient step towards $\nabla_{\mathbf{x}_{t-1}} \log p(\mathbf{c} \mid \mathbf{x}_{t-1})$ per diffusion steps. To improve performance on text and speed up decoding, we introduce two key modifications: fluency regularization and multiple gradient steps.

To generate fluent text, we run gradient updates on a control objective with *fluency regularization*: $\lambda \log p(\mathbf{x}_{t-1} \mid \mathbf{x}_t) + \log p(\mathbf{c} \mid \mathbf{x}_{t-1})$, where $\lambda$ is a hyperparameter that trades off fluency (the first term) and control (the second term). While existing controllable generation methods for diffusions do not include the $\lambda \log p(\mathbf{x}_{t-1} \mid \mathbf{x}_t)$ term in the objective, we found this term to be instrumental for generating fluent text. The resulting controllable generation process can be viewed as a stochastic decoding method that balances maximizing and sampling $p(\mathbf{x}_{t-1} \mid \mathbf{x}_t, \mathbf{c})$, much like popular text generation techniques such as nucleus sampling [14] or sampling with low temperature. In order to improve the control quality, we take multiple gradient steps for each diffusion step: we run 3 steps of the Adagrad [7] [10] update for each diffusion steps. To mitigate for the increased computation cost, we downsample the diffusion steps from 2000 to 200, which speeds up our controllable generation algorithm without hurting sample quality much.

## 5.2 Minimum Bayes Risk Decoding

Many conditional text generation tasks require a *single* high-quality output sequence, such as machine translation or sentence infilling. In these settings, we apply Minimum Bayes Risk (MBR) decoding [26] to aggregate a set of samples $\mathcal{S}$ drawn from the Diffusion-LM , and select the sample that achieves the minimum expected risk under a loss function $\mathcal{L}$ (e.g., negative BLEU score): $\hat{\mathbf{w}} = \operatorname{argmin}_{\mathbf{w} \in S} \sum_{\mathbf{w}' \in S} \frac{1}{|S|} \mathcal{L}(\mathbf{w}, \mathbf{w}')$. We found that MBR decoding often returned high quality outputs, since a low quality sample would be dissimilar from the remaining samples and penalized by the loss function.

# 6 Experimental Setup

With the above improvements on training (§4) and decoding (§5), we train Diffusion-LM for two language modeling tasks. We then apply the controllable generation method to 5 classifier-guided control tasks, and apply MBR decoding to a classifier-free control task (i.e. infilling).

## 6.1 Datasets and Hyperparameters

We train Diffusion-LM on two datasets: E2E [34] and ROCStories [32]. The E2E dataset consists of 50K restaurant reviews labeled by 8 fields including food type, price, and customer rating. The ROCStories dataset consists of 98K five-sentence stories, capturing a rich set of causal and temporal

---

[7]We tried ablations that replaced Adagrad with SGD, but we found Adagrad to be substantially less sensitive to hyperparameter tuning.

| input (Semantic Content) | food : Japanese |
| output text | Browns Cambridge is good for Japanese food and also children friendly near The Sorrento . |
| input (Parts-of-speech) | PROPN AUX DET ADJ NOUN NOUN VERB ADP DET NOUN ADP DET NOUN PUNCT |
| output text | Zizzi is a local coffee shop located on the outskirts of the city . |
| input (Syntax Tree) | (TOP (S (NP (*) (*) (*)) (VP (*) (NP (NP (*) (*)))))) |
| output text | The Twenty Two has great food |
| input (Syntax Spans) | (7, 10, VP) |
| output text | Wildwood pub serves multicultural dishes and is ranked 3 stars |
| input (Length) | 14 |
| output text | Browns Cambridge offers Japanese food located near The Sorrento in the city centre . |
| input (left context) | My dog loved tennis balls. |
| input (right context) | My dog had stolen every one and put it under there. |
| output text | One day, I found all of my lost tennis balls underneath the bed. |

Table 1: Example input control and output text for each control tasks.

commonsense relations between daily events. This dataset is more challenging to model than E2E, because the stories contain a larger vocabulary of 11K words and more diverse semantic content.

Our Diffusion-LM is based on Transformer [52] architecture with 80M parameters, with a sequence length $n = 64$, diffusion steps $T = 2000$ and a square-root noise schedule (see Appendix A for details). We treat the embedding dimension as a hyperparameter, setting $d = 16$ for E2E and $d = 128$ for ROCStories. See Appendix B for hyperparameter details. At decoding time, we downsample to 200 diffusion steps for E2E and maintain 2000 steps for ROCStories. Decoding Diffusion-LM for 200 steps is still 7x slower than decoding autoregressive LMs. For controllable generation, our method based on Diffusion-LM is 1.5x slower than FUDGE but 60x faster than PPLM.

## 6.2 Control tasks

We consider 6 control tasks shown in Table 1: the first 4 tasks rely on a classifier, and the last 2 tasks are classifier free[8]. For each control task (e.g. semantic content), we sample 200 control targets **c** (e.g., rating=5 star) from the validation splits, and we generate 50 samples for each control target. To evaluate the fluency of the generated text, we follow the prior works [55, 6] and feed the generated text to a teacher LM (i.e., a carefully fine-tuned GPT-2 model) and report the perplexity of generated text under the teacher LM. We call this metric lm-score (denoted as lm): a lower lm-score indicates better sample quality. [9] We define success metrics for each control task as follows:

**Semantic Content.** Given a field (e.g., rating) and value (e.g., 5 star), generate a sentence that covers field=value, and report the success rate by exact match of 'value'.

**Parts-of-speech.** Given a sequence of parts-of-speech (POS) tags (e.g., *Pronoun Verb Determiner Noun*), generate a sequence of words of the same length whose POS tags (under an oracle POS tagger) match the target (e.g., *I ate an apple*). We quantify success via word-level exact match.

**Syntax Tree.** Given a target syntactic parse tree (see Figure 1), generate text whose syntactic parse matches the given parse. To evaluate the success, we parse the generated text by an off-the-shelf parser [22], and report F1 scores.

**Syntax Spans.** Given a target (span, syntactic category) pair, generate text whose parse tree over span $[i, j]$ matches the target syntactic category (e.g. prepositional phrase).We quantify success via the fraction of spans that match exactly.

**Length.** Given a target length $10, \ldots, 40$, generate a sequence with a length within $\pm 2$ of the target. In the case of Diffusion-LM, we treat this as a classifier-free control task.

**Infilling.** Given a left context ($O_1$) and a right context ($O_2$) from the aNLG dataset [2], and the goal is to generate a sentence that logically connects $O_1$ and $O_2$ (algorithm details in Appendix G). For evaluation, we report both automatic and human evaluation from the Genie leaderboard [19].

---

[8]Length is classifier-free for our Diffusion-LM based methods, but other methods still require a classifier.

[9]Prior works [55, 6] use GPT [38] as the teacher LM whereas we use a fine-tuned GPT-2 model because our base autoregressive LM and Diffusion-LM both generate UNK tokens, which does not exist in pretrained vocabularies of GPT.

| | Semantic Content | | Parts-of-speech | | Syntax Tree | | Syntax Spans | | Length | |
|---|---|---|---|---|---|---|---|---|---|---|
| | ctrl ↑ | lm ↓ | ctrl ↑ | lm ↓ | ctrl ↑ | lm ↓ | ctrl ↑ | lm ↓ | ctrl ↑ | lm ↓ |
| PPLM | 9.9 | 5.32 | - | - | - | - | - | - | - | - |
| FUDGE | 69.9 | 2.83 | 27.0 | 7.96 | 17.9 | **3.39** | 54.2 | 4.03 | 46.9 | 3.11 |
| Diffusion-LM | **81.2** | **2.55** | **90.0** | **5.16** | **86.0** | 3.71 | **93.8** | **2.53** | **99.9** | **2.16** |
| FT-sample | 72.5 | 2.87 | 89.5 | 4.72 | 64.8 | 5.72 | 26.3 | 2.88 | 98.1 | 3.84 |
| FT-search | 89.9 | 1.78 | 93.0 | 3.31 | 76.4 | 3.24 | 54.4 | 2.19 | 100.0 | 1.83 |

Table 2: Diffusion-LM achieves high success rate (ctrl ↑) and good fluency (lm ↓) across all 5 control tasks, outperforming the PPLM and FUDGE baselines. Our method even outperforms the fine-tuning oracle (FT) on controlling syntactic parse trees and spans.

### 6.3 Classifier-Guided Control Baselines

For the first 5 control tasks, we compare our method with PPLM, FUDGE, and a fine-tuning oracle. Both PPLM and FUDGE are plug-and-play controllable generation approaches based on an autoregressive LM, which we train from scratch using the GPT-2 small architecture [39].

**PPLM[6].** This method runs gradient ascent on the LM activations to increase the classifier probabilities and language model probabilities, and has been successful on simple attribute control. We apply PPLM to control semantic content, but not the remaining 4 tasks which require positional information, as PPLM's classifier lacks positional information.

**FUDGE[55].** For each control task, FUDGE requires a future discriminator that takes in a prefix sequence and predicts whether the complete sequence would satisfy the constraint. At decoding time, FUDGE reweights the LM prediction by the discriminator scores.

**FT.** For each control task, we fine-tune GPT-2 on (control, text) pair, yielding an *oracle* conditional language model that's not plug-and-play. We report both the sampling (with temperature 1.0) and beam search (with beam size 4) outputs of the fine-tuned models, denoted as FT-sample and FT-search.

### 6.4 Infilling Baselines

We compare to 3 specialized baseline methods developed in past work for the infilling task.

**DELOREAN [36].** This method continuously relaxes the output space of a left-to-right autoregressive LM, and iteratively performs gradient updates on the continuous space to enforce fluent connection to the right contexts. This yields a continuous vector which is rounded back to text.

**COLD[37].** COLD specifies an energy-based model that includes fluency (from left-to-right and right-to-left LM) and coherence constraints (from lexical overlap). It samples continuous vectors from this energy-based model and round them to text.

**AR-infilling.** We train an autoregressive LM from scratch to do sentence infilling task [9]. Similar to training Diffusion-LM, we train on the ROCStories dataset, but pre-process it by reordering sentences from $(O_1, O_{\text{middle}}, O_2)$ to $(O_1, O_2, O_{\text{middle}})$. At evaluation time, we feed in $O_1, O_2$, and the model generates the middle sentence.

## 7 Main Results

We train Diffusion-LMs on the E2E and ROCStories datasets. In terms of negative log-likelihood (NLL, lower is better), we find that the variational upper bound of Diffusion-LM NLL [10] underperforms the equivalent autoregressive Transformer model (2.28 vs. 1.77 for E2E, 3.88 vs 3.05 for ROCStories) although scaling up model and dataset size partially bridges the gap (3.88 → 3.10 on ROCStories). Our best log-likelihoods required several modifications from §4; we explain these and give detailed log-likelihood results in Appendix K. Despite worse likelihoods, controllable generation based on our Diffusion-LM results in significantly better outputs than systems based on autoregressive LMs, as we will show in §7.1, §7.2, and §7.3.

### 7.1 Classifier-Guided Controllable Text Generation Results

As shown in Table 2, Diffusion-LM achieves high success and fluency across all classifier-guided control tasks. It significantly outperforms the PPLM and FUDGE baselines across all 5 tasks.

---

[10]Exact log-likelihoods are intractable for Diffusion-LM, so we report the lower bound $\mathcal{L}_{\text{vlb}}^{\text{e2e}}$.

| Syntactic Parse | ( S ( S ( NP * ) ( VP * ( NP ( NP * * ) ( VP * ( **NP ( ADJP * * ) *** ) ) ) ) ) ) * ( S ( NP * * * ) ( VP * ( ADJP ( ADJP * ) ) ) ) ) |
|---|---|
| FUDGE | Zizzi is a cheap restaurant . **[incomplete]** |
| Diffusion-LM | Zizzi is a pub providing **family friendly Indian food** Its customer rating is low |
| FT | Cocum is a Pub serving **moderately priced meals** and the customer rating is high |
| Syntactic Parse | ( S ( S ( VP * ( PP * ( NP * * ) ) ) ) * ( **NP * * *** ) ( VP * ( NP ( NP * * ) ( SBAR ( WHNP * ) ( S ( VP * ( NP * * ) ) ) ) ) ) * ) |
| FUDGE | In the city near The Portland Arms is a coffee and fast food place named The Cricketers which is not family - friendly with a customer rating of 5 out of 5 . |
| Diffusion-LM | Located on the riverside , **The Rice Boat** is a restaurant that serves Indian food . |
| FT | Located near The Sorrento, **The Mill** is a pub that serves Indian cuisine. |

Table 3: Qualitative examples from the Syntax Tree control. The syntactic parse tree is linearized by nested brackets representing the constituents, and we use the standard PTB syntactic categories. Tokens within each span are represented as * . We color failing spans red and **bold** the spans of interest that we discuss in §7.1.

| | Semantic Content + Syntax Tree | | | Semantic Content + Parts-of-speech | | |
|---|---|---|---|---|---|---|
| | semantic ctrl ↑ | syntax ctrl ↑ | lm ↓ | semantic ctrl ↑ | POS ctrl ↑ | lm ↓ |
| FUDGE | 61.7 | 15.4 | 3.52 | 64.5 | 24.1 | 3.52 |
| Diffusion-LM | **69.8** | **74.8** | 5.92 | **63.7** | **69.1** | 3.46 |
| FT-PoE | 61.7 | 29.2 | **2.77** | 29.4 | 10.5 | **2.97** |

Table 4: In this experiment, we compose semantic control and syntactic control: Diffusion-LM achieves higher success rate (ctrl ↑) at some cost of fluency (lm ↓). Our method outperforms both FUDGE and FT-PoE (product of experts of two fine-tuned models) on control success rate, especially for the structured syntactic controls (i.e. syntactic parse tree and POS).

Surprisingly, our method outperforms the fine-tuning oracle on controlling syntactic parse trees and spans, while achieving similar performance on the remaining 3 tasks.

Controlling syntactic parse trees and spans are challenging tasks for fine-tuning, because conditioning on the parse tree requires reasoning about the nested structure of the parse tree, and conditioning on spans requires lookahead planning to ensure the right constituent appears at the target position.

We observe that PPLM fails in semantic content controls and conjecture that this is because PPLM is designed to control coarse-grained attributes, and may not be useful for more targeted tasks such as enforcing that a restaurant review contains a reference to Starbucks.

FUDGE performs well on semantic content control but does not perform well on the remaining four tasks. Controlling a structured output (Parts-of-speech and Syntax Tree) is hard for FUDGE because making one mistake anywhere in the prefix makes the discriminator assign low probabilities to all continuations. In other control tasks that require planning (Length and Syntax Spans), the future discriminator is difficult to train, as it must implicitly perform lookahead planning.

The non-autoregressive nature of our Diffusion-LM allows it to easily solve all the tasks that require precise future planning (Syntax Spans and Length). We believe that it works well for complex controls that involve global structures (Parts-of-speech, Syntax Tree) because the coarse-to-fine representations allow the classifier to exert control on the entire sequence (near $t = T$) as well as on individual tokens (near $t = 0$).

**Qualitative Results.** Table 3 shows samples of Syntax Tree control. Our method and fine-tuning both provide fluent sentences that mostly satisfy controls, whereas FUDGE deviates from the constraints after the first few words. One key difference between our method and fine-tuning is that Diffusion-LM is able to correct for a failed span and have suffix spans match the target. In the first example, the generated span ("Family friendly Indian food") is wrong because it contains 1 more word than the target. Fortunately, this error doesn't propagate to later spans, since Diffusion-LM adjusts by dropping the conjunction.

## 7.2 Composition of Controls

One unique capability of plug-and-play controllable generation is its modularity. Given classifiers for multiple independent tasks, gradient guided control makes it simple to generate from the intersection of multiple controls by taking gradients on the sum of the classifier log-probabilities.

|  | Automatic Eval | | | | Human Eval |
|---|---|---|---|---|---|
|  | BLEU-4 ↑ | ROUGE-L ↑ | CIDEr ↑ | BERTScore ↑ | |
| Left-only | 0.9 | 16.3 | 3.5 | 38.5 | n/a |
| DELOREAN | 1.6 | 19.1 | 7.9 | 41.7 | n/a |
| COLD | 1.8 | 19.5 | 10.7 | 42.7 | n/a |
| Diffusion | **7.1** | **28.3** | **30.7** | **89.0** | $\mathbf{0.37}^{+0.03}_{-0.02}$ |
| AR | 6.7 | 27.0 | 26.9 | **89.0** | $\mathbf{0.39}^{+0.02}_{-0.03}$ |

Table 5: For sentence infilling, Diffusion-LM significantly outperforms prior work COLD [37] and Delorean [36] (numbers taken from paper), and matches the performance of an autoregressive LM (AR) trained from scratch to do infilling.

We evaluate this setting on the combination of Semantic Content + Syntax Tree control and Semantic Content + Parts-of-speech control. As shown in Table 4, our Diffusion-LM achieves a high success rate for both of the two components, whereas FUDGE gives up on the more global syntactic control. This is expected because FUDGE fails to control syntax on its own.

Fine-tuned models are good at POS and semantic content control individually but do not compose these two controls well by product of experts (PoE), leading to a large drop in success rates for both constraints.

### 7.3 Infilling Results

As shown in Table 5, our diffusion LM significantly outperforms continuous relaxation based methods for infilling (COLD and DELOREAN). Moreover, our method achieves comparable performance to fine-tuning a specialized model for this task. Our method has slightly better automatic evaluation scores and the human evaluation found no statistically significant improvement for either method. These results suggest that Diffusion LM can solve many types of controllable generation tasks that depend on generation order or lexical constraints (such as infilling) without specialized training.

### 7.4 Ablation Studies

We verify the importance of our proposed design choices in §4 through two ablation studies. We measure the sample quality of Diffusion-LM using the lm-score on 500 samples §6.2.

**Learned v.s. Random Embeddings (§4.1).** Learned embeddings outperform random embeddings on the ROCStories, which is a harder language modeling task. The same trend holds for the E2E dataset but with a smaller margin.

**Objective Parametrization (§4.2).** We propose to let the diffusion model predict $\mathbf{x}_0$ directly. Here, we compare this with standard

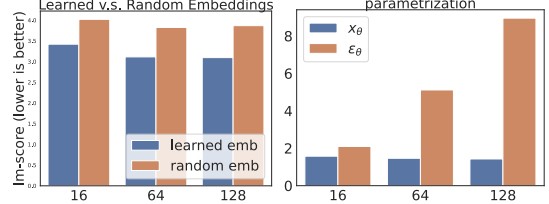

Figure 4: We measure the impact of our proposed design choices through lm-score. We find both learned embeddings and reparametrization substantially improves sample quality.

parametrization in image generation which parametrizes by the noise term $\epsilon$. Figure 4 (right) shows that parametrizing by $\mathbf{x}_0$ consistently attains good performance across dimensions, whereas parametrizing by $\epsilon$ works fine for small dimensions, but quickly collapses for larger dimensions.

## 8 Conclusion and Limitations

We proposed Diffusion-LM, a novel and controllable language model based on continuous diffusions, which enables new forms of complex fine-grained control tasks. We demonstrate Diffusion-LM's success in 6 fine-grained control tasks: our method almost doubles the control success rate of prior methods and is competitive with baseline fine-tuning methods that require additional training.

We find the complex controls enabled by Diffusion-LM to be compelling, and we are excited by how Diffusion-LM is a substantial departure from the current paradigm of discrete autoregressive generation. As with any new technologies, there are drawbacks to the Diffusion-LMs that we constructed: (1) it has higher perplexity; (2) decoding is substantially slower; and (3) training converges more slowly. We believe that with more follow-up work and optimization, many of these issues can be addressed, and this approach will turn out to be a compelling way to do controllable generation at scale.

## Acknowledgments and Disclosure of Funding

We thank Yang Song, Jason Eisner, Tianyi Zhang, Rohan Taori, Xuechen Li, Niladri Chatterji, and the members of p-lambda group for early discussions and feedbacks. We gratefully acknowledge the support of a PECASE award. Xiang Lisa Li is supported by a Stanford Graduate Fellowship.

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
