# A  Diffusion Noise Schedule

Because a diffusion model shares parameters for all diffusion steps, the noise schedule (parametrized by $\bar{\alpha}_{1:T}$) is an important hyperparameter that determines how much weight we assign to each denoising problem. We find that standard noise schedules for continuous diffusions are not robust for text data. We hypothesize that the discrete nature of text and the rounding step make the model insensitive to noise near $t = 0$. Concretely, adding small amount of Gaussian noise to a word embedding is unlikely to change its nearest neighbor in the embedding space, making denoising an easy task near $t = 0$.

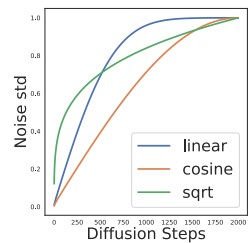

Figure 5: Visualizing the noise schedule $\sqrt{1 - \bar{\alpha}_t}$.

To address this, we introduce a new *sqrt* noise schedule that is better suited for text, shown in Figure 5 defined by $\bar{\alpha}_t = 1 - \sqrt{t/T + s}$, where $s$ is a small constant that corresponds to the starting noise level[11]. Compared to standard *linear* and *cosine* schedules, our *sqrt* schedule starts with a higher noise level and increase noise rapidly for the first 50 steps. Then *sqrt* slows down injecting noise to avoid spending much steps in the high-noise problems, which may be too difficult to solve well.

# B  Hyperparameters

**Diffusion-LM hyperparameters.**    The hyperparameters that are specific to Diffusion-LM include the number of diffusion steps, the architecture of the Diffusion-LM, the embedding dimension, and the noise schedule, . We set the diffusion steps to be 2000, the architecture to be BERT-base [7], and the sequence length to be 64. For the embedding dimensions, we select from $d \in \{16, 64, 128, 256\}$ and select $d = 16$ for the E2E dataset and $d = 128$ for ROCStories. For the noise schedule, we design the *sqrt* schedule (Appendix A) that is more robust to different parametrizations and embedding dimensions as shown in Appendix M. However, once we picked the $\mathbf{x}_0$-parametrization (§4.2) the advantage of *sqrt* schedule is not salient.

**Training hyperparameters.**    We train Diffusion-LMs using AdamW optimizer and a linearly decay learning rate starting at 1e-4, dropout of 0.1, batch size of 64, and the total number of training iteration is 200K for E2E dataset, and 800K for ROCStories dataset. Our Diffusion-LMs are trained on a single GPU: NVIDIA RTX A5000, NVIDIA GeForce RTX 3090, or NVIDIA A100. It takes approximately 5 hours to train for 200K iterations on a single A100 GPU.

To stablize the training under $\mathcal{L}_{\text{vlb}}^{\text{e2e}}$ objective, we find that we need to set gradient clipping to 1.0 and apply importance sampling to reweight each term in $\mathcal{L}_{\text{vlb}}$ [33]. Both tricks are not necessary for $\mathcal{L}_{\text{simple}}^{\text{e2e}}$ objective.

**Controllable Generation hyperparameters.**    To achieve controllable generation, we run gradient update on the continuous latents of Diffusion-LM. We use the AdaGrad optimizer [10] to update the latent variables, and we tune the learning rate, lr $\in \{0.05, 0.1, 0.15, 0.2\}$ and the trade-off parameter $\lambda \in \{0.1, 0.01, 0.001, 0.0005\}$. Different plug-and-play controllable generation approaches tradeoff between fluency and control by tunning different hyperparameters: PPLM uses the number of gradient updates per token, denoted as $k$, and we tune $k \in \{10, 30\}$. FUDGE uses the tradeoff parameter $\lambda_{\text{FUDGE}}$ and we tune this $\lambda_{\text{FUDGE}} \in \{16, 8, 4, 2\}$. Table 6 contains all the selected hyperparameter for each control tasks. Both PPLM and FUDGE has additional hyperparameters and we follow the instruction from the original paper to set those. For PPLM, we set the learning rate to be 0.04 and KL-scale to be 0.01. For FUDGE, we set precondition top-K to be 200, post top-K to be 10.

# C  Compute Cost

We run all of our experiments on a single GPU of type NVIDIA RTX A5000, NVIDIA GeForce RTX 3090, or NVIDIA A100. We report the compute cost for training and decoding Diffusion-LM.

---

[11] We set $s =$ 1e-4, and $T = 2000$, which sets the initial standard deviation to 0.1.

| | Semantic | | Parts-of-speech | | Syntax Tree | | Syntax Spans | | Length | |
|---|---|---|---|---|---|---|---|---|---|---|
| | tradeoff | lr | tradeoff | lr | tradeoff | lr | tradeoff | lr | tradeoff | lr |
| PPLM | 30 | 0.04 | - | - | - | - | - | - | - | - |
| FUDGE | 8.0 | - | 20.0 | - | 20.0 | - | 20.0 | - | 2.0 | - |
| Diffusion-LM | 0.01 | 0.1 | 0.0005 | 0.05 | 0.0005 | 0.2 | 0.1 | 0.15 | 0.01 | 0.1 |

Table 6: Hyperparameters for controllable generation methods.

| A v.s. B | A is better | B is better | similar |
|---|---|---|---|
| Diffusion-LM v.s. FUDGE | 26.7 | 8.0 | 65.3 |
| Diffusion-LM v.s. FT-sample | 24.0 | 8.0 | 68.0 |

Table 7: The human evaluation results for the semantic content control task. We report the percentage of time that our system output is more fluent, the baseline is more fluent and the two outputs have similar fluency.

**Training.**  We find that Diffusion-LM requires more training step to reach convergence than autoregressive LM. Training diffusion LM (training for 200k steps) till convergence takes around 8h on a single GPU. Training an autoregressive LM takes around 2h to converge.

**Controllable Generation.**  Controllable generation of 50 samples takes around 80s for Diffusion-LM, 4800s for (unbatched) PPLM and 50s for FUDGE.

**Decoding.**  Without control, sampling from a Diffusion-LM takes 63 seconds to obtain a batch of 50 samples. We experiment with some speed up by downsampling diffusion steps from 2000 to 200 at decoding time. The faster version takes 5.6s to obtain a batch of 50 samples. An autoregressive LM takes 0.7s to obtain a batch of 50 samples.

# D  Human Evaluation

To evaluate the output quality of Diffusion-LM, we conduct human evaluation on a subset of controllable generation experiments. We ask human to compare the outputs of Diffusion-LM and baseline methods, and report which sentence is more fluent. Out of 150 comparisons, we report the percentage of time that users prefer our system over the baselines in Table 7. The human evaluation results suggest that our system achieved high fluency and the trend is consistent with our automatic metrics of fluency, lm-score.

# E  Decoding Speed

Sampling from Diffusion-LMs requires iterating through the 2000 diffusion steps, yielding $O(2000)$ $f_\theta$ model calls. In contrast, sampling from autoregressive LMs takes $O(n)$ where $n$ is the sequence length. Therefore, decoding Diffusion-LM is slower than decoding autoregressive LMs in short and medium-length sequence regimes. Concretely, it takes around 1 minute to decode 50 sequence of length 64.

To speed up decoding, we tried skipping steps in the generative diffusion process and downsample 2000 steps to 200 steps. Concretely, we set $T = 200$ and downsample the noise schedule $\bar{\alpha}_t = \bar{\alpha}_{10t}$, which is equivalent to setting each unit transition as the transition $\mathbf{x}_t \to \mathbf{x}_{t+10}$. We decode Diffusion-LM using this new noise schedule and discretization. We find that this naive approach doesn't hurt sample quality for simple language modeling tasks like E2E, but it hurts sample quality for harder language modeling tasks like ROCStories.

For plug-and-play controllable generation tasks, extant approaches are even slower. PPLM takes around 80 minutes to generate 50 samples (without batching), because it needs to run 30 gradient updates for each token. FUDGE takes 50 seconds to generate 50 samples (with batching), because it needs to call the lightweight classifier for each partial sequence, requiring 200 classifier calls for each token, yielding $100\times$ sequence length calls. We can batch the classifier calls, but it sometimes limits batching across samples due to limited GPU memory. Our Diffusion-LM takes around 80 seconds to

| input (Semantic Content) | food : Japanese |
|---|---|
| output text | Browns Cambridge is good for Japanese food and also children friendly near The Sorrento . |
| input (Parts-of-speech) | PROPN AUX DET ADJ NOUN NOUN VERB ADP DET NOUN ADP DET NOUN PUNCT |
| output text | Zizzi is a local coffee shop located on the outskirts of the city . |
| input (Syntax Tree) | (TOP (S (NP (*) (*) (*)) (VP (*) (NP (NP (*) (*)))))) |
| output text | The Twenty Two has great food |
| input (Syntax Spans) | (7, 10, VP) |
| output text | Wildwood pub serves multicultural dishes and is ranked 3 stars |
| input (Length) | 14 |
| output text | Browns Cambridge offers Japanese food located near The Sorrento in the city centre . |
| input (left context) | My dog loved tennis balls. |
| input (right context) | My dog had stolen every one and put it under there. |
| output text | One day, I found all of my lost tennis balls underneath the bed. |

Table 8: Example input control and output text for each control tasks.

generate 50 samples (with batching). Our method downsamples the number of diffusion steps to 200, and it takes 3 classifier calls per diffusion step, yielding 600 model calls in total.

# F Classifier-Guided Controls

## F.1 Task Specification

Table 8 provides examples of the input and satisfying output pairs for each controllable generation tasks.

## F.2 Training the Classifier

**Semantic Content.** We train an autoregressive LM (GPT-2 small architecture) to predict the (field, value) pair conditioned on text. To parametrize $\log p(\mathbf{c} \mid \mathbf{x}_t)$, we compute the logprob of "value" per token.

**Parts-of-speech.** The classifier is parametrized by a parts-of-speech tagger, which estimates the probability of the target POS sequence conditioned on the latent variables. This tagger uses a BERT-base architecture: the input is the concatenated word embedding, and output a softmax distribution over all POS tags for each input word. $\log p(\mathbf{c} \mid \mathbf{x}_t)$ is the sum of POS log-probs for each word in the sequence.

**Syntax Tree.** We train a Transformer-based constituency parser [22]. Our parser makes locally normalized prediction for each span, predicting either "not a constituent", or a label for the constituent (e.g., Noun Phrase). $\log p(\mathbf{c} \mid \mathbf{x}_t)$ is the sum of log-probs for each labeled and non-constituency span in the sequence.

**Syntax Span.** We use the same parser trained for the syntax tree. $\log p(\mathbf{c} \mid \mathbf{x}_t)$ is the log-probability that the target span is annotated with the target label.

# G Infilling

Algorithm 1 shows the pseudo code for infilling tasks [30].

---

**Algorithm 1** Infilling Algorithm

---
prefix, suffix
infix length
$T$: number of diffusion steps
$p_\theta$ : trained diffusion model
$q_\theta$ : forward diffusion process

1: $x_T(\text{infix}) \sim \text{Gaussian}(0, \text{I})$
2: **for** $t \in \{T-1, \ldots, 0\}$ **do**
ß:
$\quad\quad x_{t+1}(\text{prefix}) \sim q_\theta(x_{t+1} \mid \text{EMB}(\text{prefix}))$
$\quad\quad x_{t+1}(\text{suffix}) \sim q_\theta(x_{t+1} \mid \text{EMB}(\text{suffix}))$
$\quad\quad x_{t+1} = [x_{t+1}(\text{prefix}), x_{t+1}(\text{infix}), x_{t+1}(\text{suffix})]$
$\quad\quad x_t \sim p_\theta(x_t \mid x_{t+1})$
4:
5: **return** Round($x_0(\text{infix})$)

---

## H  Non-autoregressive LMs and their controllability

Non-autoregressive (NAR) models generate all tokens in parallel, and this class of methods has been widely studied in machine translation settings [12, 27]. To improve the expressiveness of non-autoregressive models, prior works either inject latent variables [17, 29] or apply iterative refinements to generated outputs [11, 54, 50]. In particular, the iterative refinement approaches resemble discrete diffusion models [1], where the forward diffusion process randomly noises up some tokens to an absorbing state, and the backward process learns to iteratively denoise the sequence of tokens to recover a fluent sentence. The primary distinction between these two classes of methods involves the choice of noising function: simple iterative refinements relies on heuristics, whereas discrete diffusions relies on probabilitic noising function that's guaranteed to converge to some known stationary distribution.

These non-autoregressive LMs also satisfy some of these three properties that helps controllable generation (continuous, non-autoregressive, coarse-to-fine). NAR methods such as Mask-Predict are non-autoregressive and sometimes hierarchical, but often do not admit a fully continuous representation on which we can apply the gradient based control methods used in our work (in particular, Mask-Predict can be thought of as repeatedly rounding to discrete tokens). We believe studying NAR models (and discrete diffusions) in controllable generation settings would be an interesting future direction.

## I  Future Work

### I.1  Model Scaling

Scaling up model sizes has been shown to significantly improve language modeling performance [3]. For example, the sample quality significantly improves as autoregressive language models reaches the 175B parameter scale. Additionally, in the vision domain, we already see promising signs of scaling diffusions in DallE-2 [40] and Imagen [44]. Interestingly, Imagen has explored scaling laws for the diffusion U-Net and finds that scaling the diffusion parameters alone is less efficient than scaling the text encoder. We believe it's an exciting future direction to explore the efficient scaling choices for Diffusion-LMs.

### I.2  Speed-up Decoding

Currently, decoding from Diffusion-LM remains slower than autoregressive LMs. Similarly in the vision case, when DDPM came out, the generation speed was significantly slower than GANs. But some recent work has explored techniques to expedite decoding by skipping some diffusion steps at generation time [48, 23] or distilling a pre-trained diffusion model to learn a few forwarding steps [45]. We hope to adopt these advances in vision to NLP in future work.

## I.3 Speed-up Training

Slow training speed is a common problem in all diffusion models. In vision domains, diffusion models are known to be slower to train than GANs. This is because they need to learn thousands of diffusion steps jointly. We believe that speeding up training is a major open problem and exciting future work for both image and text diffusion models.

## I.4 Improving PPL

While the PPL for Diffusion-LM reported in our work is higher than the equivalently sized transformer models, this was primarily due to our focus and interest on the sample quality of the generated outputs. As with the case with image diffusions [33], we believe that techniques to optimize PPL with diffusions are likely to be different from techniques optimizing sample quality, and studing methods to bridge this PPL gap is interesting future work.

## J  End-to-end Objective Derivations

For continuous diffusion models (§3.3), $\mathcal{L}_{\text{simple}}$ is derived from the canonical objective $\mathcal{L}_{\text{vlb}}$ by reweighting each term. The first $T$ terms in $\mathcal{L}_{\text{vlb}}$ are all KL divergence between two Gaussian distributions, which has a closed form solution. Take the $t$-th term for example:

$$\mathop{\mathbb{E}}_{q(\mathbf{x}_{1:T}|\mathbf{x}_0)}\left[\log\frac{q(\mathbf{x}_{t-1}|\mathbf{x}_0,\mathbf{x}_t)}{p_\theta(\mathbf{x}_{t-1}|\mathbf{x}_t)}\right] = \mathop{\mathbb{E}}_{q(\mathbf{x}_{1:T}|\mathbf{x}_0)}\left[\frac{1}{2\sigma_t^2}||\mu_\theta(\mathbf{x}_t,t)-\hat{\mu}(\mathbf{x}_t,\mathbf{x}_0)||^2\right]+C, \qquad (3)$$

where $C$ is a constant, $\hat{\mu}$ is the mean of the posterior $q(\mathbf{x}_{t-1}|\mathbf{x}_0,\mathbf{x}_t)$, and $\mu_\theta$ is the mean of $p_\theta(\mathbf{x}_{t-1} \mid \mathbf{x}_t)$ predicted by the diffusion model. Intuitively, this simplification matches the predicted mean of $\mathbf{x}_{t-1}$ to its true posterior mean. The simplification involves removing the constant $C$ and the scaling factor $\frac{1}{2\sigma_t^2}$, yielding one term in $\mathcal{L}_{\text{simple}}$: $\mathbb{E}_{q(\mathbf{x}_{1:T}|\mathbf{x}_0)}\left[||\mu_\theta(\mathbf{x}_t,t)-\hat{\mu}(\mathbf{x}_t,\mathbf{x}_0)||^2\right]$.

To apply continuous diffusion to model discrete text, we design Diffusion-LM (§4.1) and propose a new end-to-end training objective (equation (2)) that learns the diffusion model and the embedding parameters jointly. The $\mathcal{L}_{\text{vlb}}^{\text{e2e}}$ can be written out as

$$\mathcal{L}_{\text{vlb}}^{\text{e2e}}(\mathbf{w}) = \mathop{\mathbb{E}}_{q_\phi(\mathbf{x}_0|\mathbf{w})}\left[\mathcal{L}_{\text{vlb}}(\mathbf{x}_0) + \log q_\phi(\mathbf{x}_0|\mathbf{w}) - \log p_\theta(\mathbf{w}|\mathbf{x}_0)\right]$$

$$= \mathop{\mathbb{E}}_{q_\phi(\mathbf{x}_{0:T}|\mathbf{w})}\left[\underbrace{\log\frac{q(\mathbf{x}_T|\mathbf{x}_0)}{p_\theta(\mathbf{x}_T)}}_{L_T} + \sum_{t=2}^{T}\underbrace{\log\frac{q(\mathbf{x}_{t-1}|\mathbf{x}_0,\mathbf{x}_t)}{p_\theta(\mathbf{x}_{t-1}|\mathbf{x}_t)}}_{L_{t-1}} - \underbrace{\frac{\log q_\phi(\mathbf{x}_0|\mathbf{w})}{\log p_\theta(\mathbf{x}_0|\mathbf{x}_1)}}_{L_0} - \underbrace{\log p_\theta(\mathbf{w}|\mathbf{x}_0)}_{L_{\text{round}}}\right]$$

We apply the same simplification which transforms $\mathcal{L}_{\text{vlb}} \to \mathcal{L}_{\text{simple}}$ to transform $\mathcal{L}_{\text{vlb}}^{\text{e2e}} \to \mathcal{L}_{\text{simple}}^{\text{e2e}}$:

$$\mathop{\mathbb{E}}_{q_\phi(\mathbf{x}_{0:T}|\mathbf{w})}[L_T] \to \mathbb{E}[||\mathop{\mathbb{E}}_{\mathbf{x}_T\sim q}[\mathbf{x}_T|\mathbf{x}_0]-0||^2] = \mathbb{E}[||\hat{\mu}(\mathbf{x}_T;\mathbf{x}_0)]||^2]$$

$$\mathop{\mathbb{E}}_{q_\phi(\mathbf{x}_{0:T}|\mathbf{w})}[L_{t-1}] \to \mathbb{E}[||\mathop{\mathbb{E}}_{\mathbf{x}_{t-1}\sim q}[\mathbf{x}_{t-1}|\mathbf{x}_0,\mathbf{x}_t]-\mathop{\mathbb{E}}_{\mathbf{x}_{t-1}\sim p_\theta}[\mathbf{x}_{t-1}|\mathbf{x}_t]||^2] = \mathbb{E}[||\hat{\mu}(\mathbf{x}_t,\mathbf{x}_0)-\mu_\theta(\mathbf{x}_t,t)||^2]$$

$$\mathop{\mathbb{E}}_{q_\phi(\mathbf{x}_{0:T}|\mathbf{w})}[L_0] \to \mathbb{E}[||\mathop{\mathbb{E}}_{\mathbf{x}_0\sim q_\phi}[\mathbf{x}_0\mid\mathbf{w}]-\mathop{\mathbb{E}}_{\mathbf{x}_0\sim p_\theta}[\mathbf{x}_0\mid\mathbf{x}_1]||^2] = \mathbb{E}[||\text{EMB}(w)-\mu_\theta(\mathbf{x}_1,1)||^2]$$

It's worth noting that the first term is constant if the noise schedule satisfies $\bar{\alpha}_T = 0$, which guarantees $\mathbf{x}_T$ is pure Gaussian noise. In contrast, if the noise schedule doesn't go all the way such that $\mathbf{x}_T$ is pure Gaussian noise, we need to include this regularization term to prevent the embedding from learning too large norms. Embedding with large norms is a degenerate solution, because it is impossible to sample from $p(\mathbf{x}_T)$ accurately, even though it makes all the other denoising transitions easily predictable.

Combining these terms yield $\mathcal{L}_{\text{simple}}^{\text{e2e}}$.

$$\mathcal{L}_{\text{simple}}^{\text{e2e}}(\mathbf{w}) = \mathop{\mathbb{E}}_{q_\phi(\mathbf{x}_{0:T}|\mathbf{w})} \left[ ||\hat{\mu}(\mathbf{x}_T; \mathbf{x}_0)||^2 + \sum_{t=2}^{T} [||\hat{\mu}(\mathbf{x}_t, \mathbf{x}_0) - \mu_\theta(\mathbf{x}_t, t)||^2] \right]$$
$$+ \mathop{\mathbb{E}}_{q_\phi(\mathbf{x}_{0:1}|\mathbf{w})} \left[ ||\text{EMB}(\mathbf{w}) - \mu_\theta(\mathbf{x}_1, 1)||^2 - \log p_\theta(\mathbf{w}|\mathbf{x}_0) \right].$$

Intuitively, we learn a Transformer model that that takes as input $(\mathbf{x}_t, t) \in (\mathbb{R}^{nd}, \mathbb{R})$ and the goal is to predict the distribution of $\mathbf{x}_{t-1} \in \mathbb{R}^{nd}$. It's worth noting that this Transformer model is shared across all the diffusion steps $t = 1 \ldots T$. As we demonstrated in the derivation of $\mathcal{L}_{\text{simple}}^{\text{e2e}}$, the most natural thing is to directly parametrize the neural network to predict the mean of $\mathbf{x}_{t-1} \mid \mathbf{x}_t$, we call this $\mu_\theta$-parametrization.

There are other parametrizations that are equivalent to $\mu_\theta$-parametrization up to a scaling constant. For example in §4.2, we can train the Transformer model to directly predict $\mathbf{x}_0$ via $f_\theta(\mathbf{x}_t, t)$, and use the tractable Gaussian posterior $q(\mathbf{x}_{t-1} \mid \mathbf{x}_0, \mathbf{x}_t)$ to compute the mean of $\mathbf{x}_{t-1}$, which has a closed form solution, conditioned on predicted $\mathbf{x}_0$ and observed $\mathbf{x}_t$: $\frac{\sqrt{\bar{\alpha}_{t-1}}\beta_t}{1-\bar{\alpha}_t}\mathbf{x}_0 + \frac{\sqrt{\alpha_t}(1-\bar{\alpha}_{t-1})}{1-\bar{\alpha}_t}\mathbf{x}_t$.

$$||\hat{\mu}(\mathbf{x}_t, \mathbf{x}_0) - \mu_\theta(\mathbf{x}_t, t)||^2$$
$$= ||(\frac{\sqrt{\bar{\alpha}_{t-1}}\beta_t}{1-\bar{\alpha}_t}\mathbf{x}_0 + \frac{\sqrt{\alpha_t}(1-\bar{\alpha}_{t-1})}{1-\bar{\alpha}_t}\mathbf{x}_t) - (\frac{\sqrt{\bar{\alpha}_{t-1}}\beta_t}{1-\bar{\alpha}_t}f_\theta(\mathbf{x}_t, t) + \frac{\sqrt{\alpha_t}(1-\bar{\alpha}_{t-1})}{1-\bar{\alpha}_t}\mathbf{x}_t)||^2$$
$$= ||\frac{\sqrt{\bar{\alpha}_{t-1}}\beta_t}{1-\bar{\alpha}_t}(\mathbf{x}_0 - f_\theta(\mathbf{x}_t, t))||^2$$
$$\propto ||\mathbf{x}_0 - f_\theta(\mathbf{x}_t, t)||^2$$

These two parametrizations differ by a constant scaling, and we apply the $\mathbf{x}_0$-parametrization to all terms in $\mathcal{L}_{\text{simple}}^{\text{e2e}}$ to reduce rounding errors as discussed in §4.2:

$$\mathcal{L}_{\mathbf{x}_0\text{-simple}}^{\text{e2e}}(\mathbf{w}) = \mathop{\mathbb{E}}_{q_\phi(\mathbf{x}_{0:T}|\mathbf{w})} \left[ ||\hat{\mu}(\mathbf{x}_T; \mathbf{x}_0)||^2 + \sum_{t=2}^{T} [||\mathbf{x}_0 - f_\theta(\mathbf{x}_t, t)||^2] \right]$$
$$+ \mathop{\mathbb{E}}_{q_\phi(\mathbf{x}_{0:1}|\mathbf{w})} \left[ ||\text{EMB}(\mathbf{w}) - f_\theta(\mathbf{x}_1, 1)||^2 - \log p_\theta(\mathbf{w}|\mathbf{x}_0) \right].$$

To generate samples from a Diffusion-LM with $\mathbf{x}_0$-parametrization, at each diffusion step, the model estimates the $\mathbf{x}_0$ via $f_\theta(\mathbf{x}_t, t)$ and then we sample $\mathbf{x}_{t-1}$ from $q(\mathbf{x}_{t-1} \mid f_\theta(\mathbf{x}_t, t), \mathbf{x}_t)$, which is fed as input to the next diffusion step.

# K   Log-Likelihood Models and Results

To investigate Diffusion-LM's log-likelihood performance, we make several departures from the training procedure of §4. Ultimately the log-likelihood improvements described in this section did not translate into better generation quality in our experiments and therefore we focus on the original method in the rest of the paper. Our likelihood models are trained as follows:

- Instead of training a diffusion model on sequences of low-dimensional token embeddings, we train a model directly sequences of on one-hot token vectors.
- Following the setup of Kingma et al. [21], we train a continuous-time diffusion model against the log-likelihood bound and learn the noise schedule simultaneously with the rest of the model to minimize the loss variance.
- Because our model predicts sequences of one-hot vectors, we use a softmax nonlinearity at its output and replace all squared-error terms in the loss function with cross-entropy terms. This choice of surrogate loss led to better optimization, even though we evaluate against the original loss with squared-error terms.
- The model applies the following transformation to its inputs before any Transformer layers: $x := \text{softmax}(\alpha(t)x + \beta(t))$ where $\alpha(t) \in \mathbb{R}$ and $\beta(t) \in \mathbb{R}^v$ are learned functions of the diffusion timestep $t$ parameterized by MLPs ($v$ is the vocabulary size).

| Dataset | Small AR | Small Diffusion | Medium Diffusion |
|---|---|---|---|
| E2E | 1.77 | 2.28 | - |
| ROCStories | 3.05 | 3.88 | - |
| ROCStories (+GPT-J) | 2.41 | 3.59 | 3.10 |

Table 9: Log-likelihood results (nats per token)

- At inference time, we omit the rounding procedure in §4.2.

For exact model architecture and training hyperparameter details, please refer to our released code.

We train these diffusion models, as well as baseline autoregressive Transformers, on E2E and ROCStories and report log-likelihoods in Table 9. We train two sizes of Transformers: "small" models with roughly 100M parameters and "medium" models with roughly 300M parameters. Both E2E and ROCstories are small enough datasets that all of our models reach their minimum test loss early in training (and overfit after that). To additionally compare model performance in a large-dataset regime, we also present "ROCStories (+GPT-J)" experiments in which we generate 8M examples of synthetic ROCStories training data by finetuning GPT-J [53] on the original ROCStories data, pretrain our models on the synthetic dataset, and then finetune and evaluate them on the original ROCStories data.

## L   Qualitative Examples

We show randomly sampled outputs of Diffusion-LM both for unconditional generation and for the 5 control tasks. Table 10 shows the unconditional generation results. Table 11, Table 12, Table 14, and Table 3 show the qualitative samples from span control, POS control, semantic content control, and syntax tree control, respectively. Table 13 shows the results of length control.

## M   Additional Ablation Studies

In addition to the 2 ablation studies in §7.4, we provide more ablation results in Figure 6 about architecture choices and noise schedule.

**Learned v.s. Random Embeddings (§4.1).** Learned embeddings outperform random embeddings on both ROCStories and the E2E dataset by xx percent and xx percent respectively, as shown in the first row of Figure 6.

**Noise Schedule (Appendix A).** We compare the *sqrt* schedule with *cosine* [33] and *linear* [13] schedules proposed for image modeling. The middle row of Figure 6 demonstrates that *sqrt* schedule attains consistently good and stable performance across all dimension and parametrization choices. While the *sqrt* schedule is less important with $\mathbf{x}_0$-parametrization, we see that it provides a substantially more robust noise schedule under alternative parametrizations such as $\epsilon$.

**Transformer v.s. U-Net.**   The U-Net architecture in Ho et al. [13] utilizes 2D-convolutional layers, and we imitate all the model architectures except changing 2D-conv to 1D-conv which is suitable for text data. Figure 6 (last row) shows that the Transformer architecture outperforms U-Net.

## N   Societal Impacts

On the one hand, having strong controllability in language models will help with mitigating toxicity, making the language models more reliable to deploy. Additionally, we can also control the model to be more truthful, reducing the inaccurate information generated by the language model by carefully controlling it to be truthful. On the other hand, however, one could also imagine more powerful targeted disinformation (e.g., narrative wedging) derived from the fine-grained controllability.

Towards this end, it might be worth considering generation methods that can watermark the generated outputs without affecting its fluency, and this type of watermark could also be framed as a controllable generation problem, with distinguish-ability and fluency as the constraints.

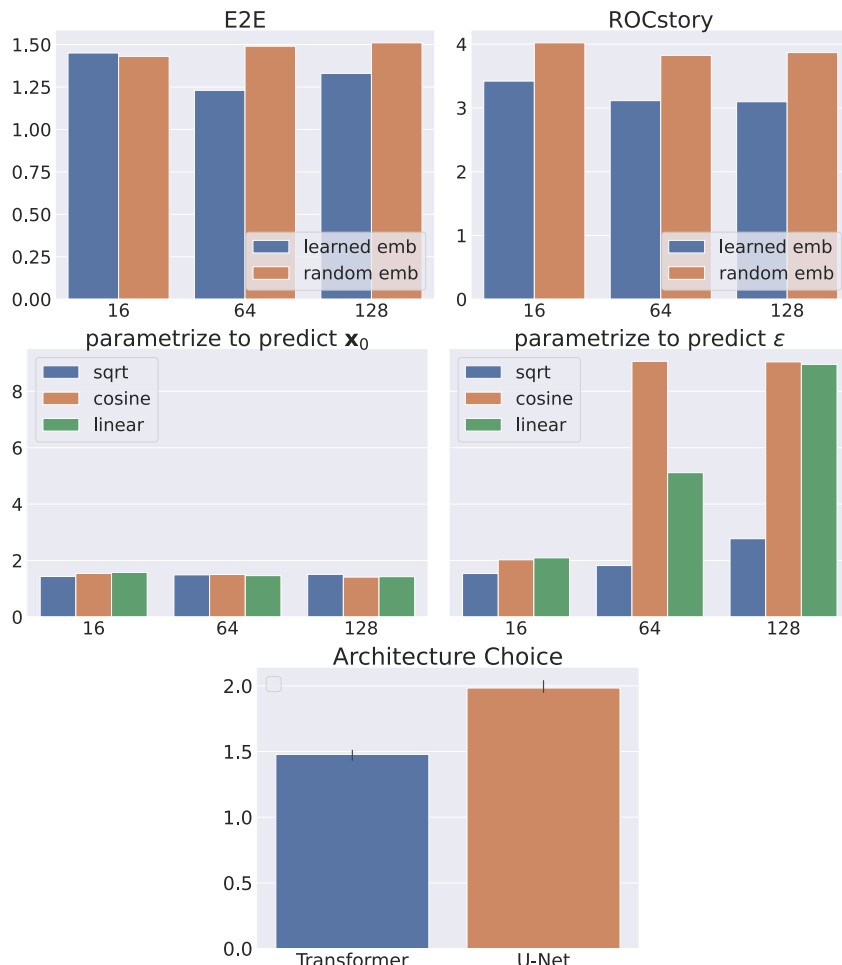

Figure 6: Additional ablation results. The first row shows Diffusion-LM with trainable embeddings outperform random embeddings on both datasets (§4.1). The second row demonstrates that *sqrt* schedule attains consistently good and stable performance across all dimension and parametrization choices. The last row shows that Transformer architecture outperforms U-Net architecture for language modeling.

| | |
|---|---|
| ROCStories+Aug | Matt was at the store . He was looking at a new toothbrush . He found the perfect one . When he got home , he bought it . It was bright and he loved it . |
| | I and my friend were hungry . We were looking for some to eat . We went to the grocery store . We bought some snacks . We decided to pick up some snacks . |
| | I was at the store . I had no money to buy milk . I decided to use the restroom . I went to the register . I was late to work . |
| | The man wanted to lose weight . He did n't know how to do weight . He decided to start walking . He ate healthy and ate less . He lost ten pounds in three months . |
| | I went to the aquarium . I wanted to feed something . I ordered a fish . When it arrived I had to find something . I was disappointed . |
| ROCStories | Tom was planning a trip to California . He had fun in the new apartment . He was driving , until it began to rain . Unfortunately , he was soaked . Tom stayed in the rain at the beach . |
| | Carrie wanted a new dress . She did not have enough money . She went to the bank to get one , but saw the missed . Finally , she decided to call her mom . She could not wait to see her new dress . |
| | Tina went to her first football game . She was excited about it . When she got into the car she realized she forgot her hand . She ended up getting too late . Tina had to start crying . |
| | Michael was at the park . Suddenly he found a stray cat . He decided to keep the cat . He went to his parents and demanded a leg . His parents gave him medicine to get it safe . |
| | Tim was eating out with friends . They were out of service . Tim decided to have a pizza sandwich . Tim searched for several hours . He was able to find it within minutes . |
| E2E | The Waterman is an expensive pub that serves Japanese food . It is located in Riverside and has a low customer rating . |
| | A high priced pub in the city centre is The Olive Grove . It is a family friendly pub serving French food . |
| | The Rice Boat offers moderate priced Chinese food with a customer rating of 3 out of 5 . It is near Express by Holiday Inn . |
| | There is a fast food restaurant , The Phoenix , in the city centre . It has a price range of more than Ŏ0a3 30 and the customer ratings are low . |
| | The Mill is a coffee shop based in the city centre area near to The Sorrento . It is in the high price range and serves Indian food . |

Table 10: Randomly sampled examples generated by unconditional sampling Diffusion-LM trained on 3 datasets. ROCStories+Aug denotes ROCStories with data augmentation. It's generated by first fine-tuning GPT-j on the ROCStories dataset and then sample the large GPT-j model to generate 1M stories.

| target span | [3, 5, PP] |
|---|---|
| FUDGE | UNK the UNK for Italian food , The Eagle coffee shop is near Burger King in the riverside area . The Eagle has a customer rating of 5 out of 5 , and isn ' t family - friendly . The Eagle has a cheap price range . |
| Diffusion-LM | The Plough , near Café Rouge , is a high priced fast food pub . |
| FT | Along the riverside near Café Rouge is The Golden Curry . It serves Italian food in a family - friendly environment . It has a low customer rating . |

| target span | [10, 12, PP] |
|---|---|
| FUDGE | Blue Spice is a high price range Fast food located in city centre . |
| Diffusion-LM | The Phoenix is a high priced food restaurant , located near the river . |
| FT | The Punter is a family restaurant with low prices and delicious sushi , located near the Café Sicilia |

| target span | [9, 14, S] |
|---|---|
| FUDGE | Zizzi pub serves Italian food for adults only . It has been rated average by customers . |
| Diffusion-LM | There is a Chinese restaurant called The Eagle , it has an average customer rating . |
| FT | On the riverside area are located Alimentum , has a very good French food for adults and kids , UNK price range are over 20 to 25 £ . |

| target span | [4, 16, VP] |
|---|---|
| FUDGE | The Cambridge Blue pub is near the Café Brazil and offers a high price range for their French food . |
| Diffusion-LM | On the Ranch there is a children friendly pub called The Cricketers with an average customer rating . |
| FT | The Travellers Rest Beefeater is an average rated restaurant located in the riverside area near Café Adriatic . Their price range is less than £ 20 . |

| target span | [0, 2, NP] |
|---|---|
| FUDGE | The Golden Palace is a cheap , 5 - star coffee shop , located on the river in the north of the city centre . |
| Diffusion-LM | The Olive Grove is a pub that provides Indian food in the high price range . It is in the city centre . |
| FT | The Golden Curry is located in city centre near Café Rouge which provides English food . Its customer rating is average and is not family - friendly . |

| target span | [12, 13, NP] |
|---|---|
| FUDGE | The Waterman is a family friendly place with a good rating .[missing span] |
| Diffusion-LM | The Vaults is a high priced , family friendly restaurant that serves Italian food . |
| FT | Strada is a restaurant which costs less than £ 20 , but is not family - friendly and has an average rating . |

Table 11: Qualitative output of the syntax span control tasks. The target span $(i, j, \text{label})$ means the span from position $i$ to position $j$ should be a constituent with a specific label: S is sentence, NP is noun phrase, VP is verb phrase, PP is prepositional phrase, etc. We color failed spans red and correct spans green.

| | |
|---|---|
| target POS | PROPN AUX DET ADJ NOUN NOUN VERB ADP DET NOUN ADP DET NOUN PUNCT |
| FUDGE | Aromi is a non family - friendly fast food coffee shop in the riverside area with a low Customer Rating . |
| Diffusion-LM | Fitzbillies is a cheap coffee shop located on the outskirts of the city . |
| FT | Aromi is a fast food pub located at the centre of the city. |
| target POS | PROPN AUX DET NOUN VERB NOUN ADJ NOUN PUNCT PRON NOUN NOUN AUX ADJ |
| FUDGE | Cocum is a family - friendly coffee shop , that has a low price range and a low customer rating . |
| Diffusion-LM | Zizzi is a pub providing restaurant Chinese food . Its customer rating is low |
| FT | Zizzi is a pub providing kids friendly services. Its customer rating is average |
| target POS | DET NOUN PUNCT PROPN VERB ADJ CCONJ ADJ NOUN CCONJ AUX VERB ADP DET PROPN ADJ PROPN PUNCT |
| FUDGE | A child - friendly coffee shop , Cocum , offers fast food at an average price range of £ 20 - 25 . |
| Diffusion-LM | The Waterman - friendly serves UNK and fast food and is located near the Crown Plaza Hotel . |
| FT | The wine - Strada serves fast and cheap food and is located near the Rainbow Vegetarian Café. |
| target POS | DET PROPN PROPN VERB ADJ NOUN ADP NOUN ADP SYM NUM PUNCT NOUN NOUN PROPN VERB ADP DET PROPN CCONJ PROPN ADP PROPN PROPN PUNCT ADJ PUNCT DET NOUN PART AUX VERB PUNCT |
| FUDGE | The Midsummer House offers cheap English food near All Bar One . Rated 5 out of 5 . |
| Diffusion-LM | The Rice Boat provides Chinese food in £ 20 - 25 . Price range is high . The Rice Boat is located near the Express by Holiday Inn and is kids friendly . The customer rating is high . |
| FT | The Rice Boat welcomes Japanese food with prices under £ 20. Customer ratings are low. The Rice Boat is located near the Express by Holiday Inn. Convenient. No children's are allowed. |
| target POS | PROPN PROPN AUX DET ADJ NOUN NOUN ADP DET NOUN NOUN ADP DET PROPN PUNCT PRON AUX NOUN PUNCT ADJ PUNCT |
| FUDGE | Loch Fyne is a Japanese restaurant with a moderate price range and kid - friendly atmosphere . |
| Diffusion-LM | Browns Cambridge is an Italian restaurant shop in the city centre near The Sorrento . It is family - friendly . |
| FT | Browns Cambridge is a cheap coffee shop in the riverside area near The Sorrento, that is family - friendly. |
| target POS | PROPN VERB DET ADJ NOUN NOUN PROPN PUNCT PRON AUX ADJ VERB CCONJ VERB NOUN SCONJ VERB ADJ NOUN PUNCT |
| FUDGE | Fitzbillies coffee shop has a high price range , children friendly service and serves Japanese food in riverside with high customer rating . |
| Diffusion-LM | There has a high customer rating . It is kid friendly called The Golden Curry and serves Indian food . |
| FT | Customers give the French coffee shop Fitzbillies ; it is average rated and offers families where serving light meals. |
| target POS | DET NUM NUM VERB ADJ NOUN |
| FUDGE | The Twenty Two serves Fast food and is kids friendly . |
| Diffusion-LM | The Twenty Two provides Chinese food |
| FT | The Twenty Two provides Indian food |
| target POS | ADV NOUN ADV ADP PROPN PROPN PUNCT DET PROPN NOUN NOUN VERB ADJ NOUN NOUN CCONJ AUX PART VERB NOUN NOUN PUNCT |
| FUDGE | UNK your whole family to The Wrestlers , the best UNK the UNK UNK at the river |
| Diffusion-LM | Located in riverside near The Sorrento , Browns Cambridge coffee shop serves Japanese food , and is not family - friendly . |
| FT | Even adults only at Loch Fyne, The Rice Boat coffee shop has moderate price range and does not cater kids age. |
| target POS | DET PROPN AUX DET NUM NOUN NOUN NOUN VERB ADP DET PROPN PROPN PUNCT |
| FUDGE | The Eagle is a 3 star coffee shop located near Burger King , north of the City centre that provides low - cost fast food . |
| Diffusion-LM | The Cricketers is a five star coffee shop located near The Portland Arms . |
| FT | The Vaults is a one star coffee shop located near the Café Brazil. |

Table 12: Qualitative output of the POS control tasks. The target POS is the sequence of gold parts-of-speech tags the generated texts should match.

| target length | 7 |
|---|---|
| FUDGE | Wildwood is a cheap Japanese pub . Low rating . |
| Diffusion-LM | The Twenty Two serves Indian food . |
| FT | The Mill is an Indian restaurant . |

| target length | 12 |
|---|---|
| FUDGE | The Phoenix is an average Japanese restaurant that is in the City Centre . |
| Diffusion-LM | The Twenty Two serves Chinese food and is not family friendly . |
| FT | Green Man is an average priced restaurant located near All Bar One |

| target length | 17 |
|---|---|
| FUDGE | Fitzbillies is an expensive Italian coffee shop in the city centre . It is not child friendly. . |
| Diffusion-LM | The Twenty Two serves Indian food in the city centre . It is not family friendly . |
| FT | For low - priced food and a family - friendly atmosphere, visit Fitzbillies near Express by Holiday Inn |

| target length | 22 |
|---|---|
| FUDGE | The Golden Curry is an English food restaurant located near the Café Rouge in the Riverside area . The customer rating is average . Children are welcome . |
| Diffusion-LM | Strada is a fast food pub located near Yippee Noodle Bar and has a customer rating of 3 out of 5 . |
| FT | There is an Italian kid friendly restaurant in the riverside area near The Sorrento named Browns Cambridge in the riverside area . |

| target length | 27 |
|---|---|
| FUDGE | The Olive Grove is an expensive , children friendly , Fast food restaurant in the city centre . [missing 9 words] |
| Diffusion-LM | The Eagle is a family friendly coffee shop in the city centre near Burger King . It serves Italian food and has a low customer rating . |
| FT | A pub in the city centre near Yippee Noodle Bar is named Strada. It serves French food and has a customer rating of 3 out of 5 |

| target length | 32 |
|---|---|
| FUDGE | The Golden Curry is a Japanese food restaurant with a high customer Rating , kid friendly and located along the riverside near Café Rouge . [missing 7 words] |
| Diffusion-LM | There is a family - friendly coffee shop in the city centre , it is called Zizzi . It is cheap and has a customer rating of 5 out of 5 . |
| FT | In the city centre is a kids friendly place called Green Man. It has Japanese food and is near All Bar One. It has a price range of £ 20 - 25 |

| target length | 37 |
|---|---|
| FUDGE | There is a coffee shop called Fitzbillies which offers French food at cheap prices . It is not family - friendly and has a customer rating of 5 out of 5 . It is in riverside . |
| Diffusion-LM | The Rice Boat provides Indian food in the moderate price range . It is located in the city centre . It is near Express by Holiday Inn . Its customer rating is 3 out of 5 . |
| FT | For a family friendly coffee shop that serves Italian food, with a customer rating of 5 out of 5 and a cheap price range, try The Eagle. It is located in the riverside area . |

Table 13: Qualitative output of the length control tasks, where all the generated texts tried to exactly match the target length. We mark the words exceeding the target length red.

| target semantic content | name : Bibimbap House |
|---|---|
| FUDGE | Clare Hall , the Bibimbap House , serves high end Japanese food in the city centre . |
| Diffusion-LM | Bibimbap House in riverside near Clare Hall has a cheap price range . |
| FT | By Clare Hall is Bibimbap House which serves expensive noodles. |

| target semantic content | name : Travellers Rest Beefeater |
|---|---|
| FUDGE | Clowns near Clare Hall in riverside is a French coffee shop rated 5 out of 5 |
| Diffusion-LM | Green Man is an Italian pub located in the city centre near Café UNK . |
| FT | Travellers Rest Beefeater is a reasonably priced restaurant that is family friendly. |

| target semantic content | Type : coffee shop |
|---|---|
| FUDGE | Wildwood is a coffee shop located near Ranch . It is expensive and highly UNK . |
| Diffusion-LM | The Punter is a high priced coffee shop located near Café Sicilia that serves Japanese food . It is not family - friendly and has a customer rating of 3 out of 5 . |
| FT | Located in the riverside area is the coffee shop Fitzbillies. It has Indian food in the price Range of less than £ 20 and a low customer Rating. It is not family Friendly. |

| target semantic content | customer rating : low |
|---|---|
| FUDGE | The Waterman is a fast food restaurant that is family - friendly near the city centre . [missing content] |
| Diffusion-LM | The Rice Boat restaurant has a low customer rating and is located in riverside . It serves Italian food , and is not family - friendly . |
| FT | The Eagle is low customer rating coffee shop with Italian food in riverside near Burger King. Its price range is less than £ 20 and is family - friendly. |

| target semantic content | near : The Sorrento |
|---|---|
| FUDGE | Browns Cambridge provides Indian food in the less than £ 20 price range . Its customer rating is low . [missing content] |
| Diffusion-LM | Near The Sorrento on the riverside is a pub named Taste of Cambridge that serves Japanese food . |
| FT | Browns Cambridge sells Italian food and is also a coffee shop. It has an average customer rating. It is located in the riverside area near Crowne Plaza Hotel and yes it is child friendly. |

| target semantic content | food : Italian |
|---|---|
| FUDGE | A non family - friendly Italian pub is Zizzi . It has an average customer rating . |
| Diffusion-LM | Loch Fyne is Italian Japanese restaurant that is kid friendly . |
| FT | situated near All Bar One is a child friendly Italian eatery called Green Man costing more than £ 30 is a restaurant near the riverside |

| target semantic content | price : high |
|---|---|
| FUDGE | The Vaults is a high priced Italian Pub with a customer rating of 3 out of 5 near Café Adriatic |
| Diffusion-LM | The Punter is a French restaurant with a high price range . |
| FT | A fast food coffee shop that is not kid friendly is called Cocum. It is expensive and gets average ratings. |

Table 14: Qualitative output of the semantic content control task. We mark the compliant spans as green, and the spans that violates the control target as red.