# OpenReview forum: "Diffusion-LM Improves Controllable Text Generation"
_NeurIPS.cc/2022/Conference — NeurIPS 2022 Accept_

### Official Review · Reviewer_Pvpw · 2022-07-07

**Rating:** 7
**Confidence:** 4
**Soundness:** 4 excellent
**Presentation:** 4 excellent
**Contribution:** 3 good

**Summary:**

This paper proposed Diffusion-LM, a non-autoregressive language model based on continuous diffusions, for controllable text generation. The non-autoregressive model would iteratively denoise a sequence of Gaussian vectors into word vectors, and the control information is injected by running gradient updates on intermediate latent variables. The main contribution of this paper is that it demonstrated how to combine non-autoregressive model and continuous diffusion model and use it for controllable text generation. The experiment result shows that Diffusion-LM significantly outperforms prior work.

**Questions:**

1. I'm not sure about the training process, is this understanding right: the output vectors of each decoding pass of the non-autoregressive model are called latent variables, and these variables are used to predict the control information. After the diffusion process, the final output  would compute a cross entropy loss with the reference.

2. I don't see the reason why evaluating on syntax-related tasks. Are these tasks useful in real scenario? Or they are just deigned tasks to highlight the strengths of Diffusion-LM?

**Limitations:**

The authors had mentioned their limitations in the paper: higher PPL and slower training, and the training process itself might need large GPU memory. The fewer diffusion steps might speed up decoding process but needs further exploration to avoid performance loss.

**Strengths And Weaknesses:**

Strengths:
1. The writing is good and the paper is well presented.
2. Detailed derivations are included, and the experiments are adequate and demonstrate good performance.
3. Combining the diffusion model from CV and non-autoregressive model is not a trivial task, this paper explores the changes and improvements needed to apply Diffusion-LM on controllable text generation.

Weakness:
1. The training speed is much slower and the generated text suffers from higher PPL.
2. Some downstream tasks still need three classifiers and the cost for these classifiers is high.
3. Some downstream tasks may lack practical application.

---

> ### Author Response · Authors · 2022-08-02
> **Response to Reviewer Pvpw**
>
> Dear reviewer, thank you for the insightful comments. We appreciate that you recognize our technical contribution to adapt diffusion models to language data, and we are glad you like our paper. We address your comments below.
>
> ----
> [Weakness 1]
>
> Slow training speed is a common problem in all diffusion models. In vision domains, diffusion models are known to be slower to train than GANs. This is because they need to learn 2000 diffusion steps jointly. The main contribution of this paper is to introduce the new language model class (Diffusion-LM) and provide initial evidence that diffusion models also work for language data. We believe that speeding up training is a major open problem and exciting future work for both image and text diffusion models. Given recent successes and interest in scaling image diffusions, as well as improvements to decoding speed (DALLE-2 https://cdn.openai.com/papers/dall-e-2.pdf; Imagen https://arxiv.org/pdf/2205.11487.pdf;  DDIM https://arxiv.org/abs/2010.02502;), we are optimistic that further work along these lines will improve this drawback over time.
>
> While the PPL for diffusion-lm reported in our work is higher than the equivalently sized transformer models, this was primarily due to our focus and interest on the sample quality of the generated outputs. As with the case with image diffusions (Improved Denoising Diffusion Probabilistic Models https://arxiv.org/abs/2102.09672 ), we believe that techniques to optimize PPL with diffusion are likely to be different from techniques optimizing sample quality, and we will study methods to bridge this PPL gap in follow-up studies.
>
> ----
>
> [Weakness 2]
> For experiments in section 6.1 (Table 1), we need to train one classifier for each control task (for example, controlling POS taggings require a single POS tagger as the classifier. Controlling the attributes requires a simple generator that takes in (restaurant review, field) and outputs the value of that field in the review.)  For experiments in section 6.2 (Table 3),  we evaluate the composition capacity of Diffusion-LM, by reusing classifiers trained in section 6.1.
> Also, these classifiers used for Diffusion-LM are quite easy to train and converge quickly. It takes around 1h on a single GPU to train each classifier. Other plug-and-play controllable generation baselines (FUDGE, PPLM) also require training a classifier for each task. Overall, the cost of training these classifiers is lower than fine-tuning the language model for the task, and can sometimes achieve better performance (especially in composition).
>
> ----
>
> [Weakness 3]
> The reviewer mentioned that some downstream tasks may lack practical application. The downstream tasks in our experiments include semantic content control, length control, and infilling, which we believe to have direct practical applications. We agree that control tasks like POS tags or syntax tree controls are quite linguistics-focused and don't have direct applications. However, there are some practical tasks in sentence simplification that control syntactic trees as an intermediate step.  https://arxiv.org/pdf/1910.04387.pdf  and https://aclanthology.org/W13-2901.pdf
>
> ----
>
> Thanks for the questions:
>
> [Question 1]
> We interpreted this question as asking for the method we use for controllable generation. Please let us know if we have misinterpreted it.
>
> Controllable text generation only happens after Diffusion-LM is trained.
> Once we have a trained Diffusion-LM, the output vector of the at the t-th diffusion step is the predicted mean of the latent variable $x_{t-1}$.  For unconditional decoding, we sample $x_{t-1}$ from this distribution $p( x_{t-1} | x_t)$. For controllable generation, we first sample $x_{t-1}$ via the diffusion process (above) and then update $x_{t-1}$ via gradient steps on $\log p($ control$ | x_{t-1})$).
>
> ----
>
> [Question 2]
> We choose syntactic control in this task to stress the strength of our method, as we consider controlling the entire syntax tree to be a complicated task, addressing both high-level syntactic planning and low level lexical controls. Being able to deal with syntax trees with high success rates could be strong evidence that our method is able to handle complex and fine-grained control tasks.

---

### Official Review · Reviewer_sK4s · 2022-07-11

**Rating:** 6
**Confidence:** 3
**Soundness:** 3 good
**Presentation:** 4 excellent
**Contribution:** 4 excellent

**Summary:**

This paper introduces a new non-autoregressive language model based on continuous diffusion models, which have been quite successful in the vision area. Specifically, this paper proposes a novel training objective to train an embedding function that can convert the discrete text tokens into a continuous space. Correspondingly, this paper also introduces methods to improve the rounding from a continuous latent variable to discrete text. The abovementioned techniques facilitate training a Diffusion-LM. Besides, this paper shows that the control over text generation can be done on continuous latent space and the proposed approach yields successful control of Diffusion-LM for six fine-grained control tasks compared to prior work.

**Questions:**

- Line 77-78, I don't think the cited paper has shown that current non-autoregressive approaches fail for language modeling. Would you provide some more details here?
- Line 174, I do not quite understand the following sentence:
" a mixture of Dirac delta distribution will only appear in the terms with $t$ near zero". Could you elaborate more?
- In addition to the fine-grained text control tasks, I wonder how the proposed model performs on coarse-grained tasks, e.g., controlling simple sentence attributes like sentiment?


**Limitations:**

Yes

**Strengths And Weaknesses:**

**Strengths**
- Very well-written, I enjoyed reading the paper
- There have been few works exploring diffusion models on NLP and this is the first work to explore continuous diffusion models on NLP. It looks definitely interesting and could potentially inspire others to keep working along this line.
- The proposed end-to-end training looks effective in learning embeddings.
- Empirical results demonstrate the success of text control for *fine-grained* control tasks.


**Weakness**
- My primary concern is the scalability of Diffusion-LM. Though under the current experiment setup, i.e., 80M parameters for Diffusion-LM and GPT-2 Small, the proposed approach outperforms existing methods based on autoregressive LM on text control. One of the advantages of existing works is that they can use arbitrary large pre-trained LMs that are available. It is not clear to me how the scaling low works for the current proposed Diffusion-LM and text control approach.
- Though Diffusion-LM achieves better performance on text control, I wonder what could be the most appealing reason to choose a Diffusion-LM over an autoregressive LM generally? As I understand, autoregressive LMs can benefit many downstream NLP tasks in addition to the text control. I am curious about what justifies the application of Diffusion-LM here.

---

> ### Author Response · Authors · 2022-08-02
> **Response to Reviewer sK4s**
>
> Dear reviewer, thank you for the insightful comments. We are glad that you enjoy reading our paper and recognize the novelty and impact of our work. We address your comments below.
>
> ----
> [weakness 1]
> The main contribution of this paper is to introduce the new language model class (Diffusion-LM) and develop techniques that enable diffusion models to work for language data. While building Diffusions of comparable scale to autoregressive models would be ideal, we do not have access to compute resources that would enable us to brute-force this problem, and we believe that further follow-up work (including our own) will substantially improve efficiencies of Diffusion-LMs in the coming years. In the vision domain, we already see promising signs of scaling diffusions in DallE-2 and Imagen.
>
> That said, we acknowledge the importance of scale to high-performance language modeling, and we have added additional discussions in Appendix I about scaling laws in diffusion models for vision (DallE-2, Imagen), and how this might guide scaling choices for future Diffusion-LMs.
>
> ----
> [Weakness 2]
> Other than controllable generation, Diffusion-LM enables flexible factorization, and generates text with arbitrary orderings. This flexible generation ordering can facilitate many practical applications like writing assistants, code infilling completion. Admittedly, the Diffusion-LM are much smaller than existing GPT* language models, which is the primary reason people would choose AR models over Diffusion-LM. We hope this work would encourage more future research that scales Diffusion-LM to improve its generation capacities.
>
> ----
> Thanks for the questions:
>
> [Question 1]  Based on our literature search, most non-autoregressive models are trained and evaluated only on machine translation tasks. For example:
> Mask-Predict: Parallel Decoding of Conditional Masked Language Models https://arxiv.org/pdf/1904.09324.pdf
> Non-Autoregressive Neural Machine Translation https://arxiv.org/pdf/1711.02281.pdf
> Deterministic Non-Autoregressive Neural Sequence Modeling by Iterative Refinement https://aclanthology.org/D18-1149/
> A Study of Non-autoregressive Model for Sequence Generation https://aclanthology.org/2020.acl-main.15.pdf
>
> The paper we cited analyzed why NAR methods would work for some NMT tasks but not others, and provided justification by measuring target word dependence. We think open-ended generation requires strong target word dependence, which would be more challenging for NAR models. To make the writing clear, we will add discussions that using NAR models for open-ended LM remains an open problem in the appendix H.
>
> ----
>
> [Question 2] Diffusion steps with t near 0 are very close to the noiseless x_0. Since $p(x_0 \mid w)$ is defined by very spiky Gaussian centered around the sequence of word embeddings $N(Emb(w), \sigma_0 I )$, for small t, the distribution $p(x_t \mid w)$ is also very spiky Gaussians. Thanks for pointing this out: we use “nearly a Dirac Delta” to refer to low-variance Gaussian. An exact Dirac Delta would only happen if we select $\sigma_0$ to be exact 0. Empirically, we select $\sigma_0$ to be 0.0001.
>
> ----
>
> [Question 3] In preliminary experiments, we run the experiment to control sentiment, and our method is able to achieve a high success rate on par with FUDGE. We didn't include this experiment into the paper, because we want to focus on complex control tasks, which could be more practically useful than attribute control and better demonstrate the strength of Diffusion-LM.

---

> > ### Comment · Reviewer_sK4s · 2022-08-08
> > **Thanks for the responses**
> >
> > Thank you for addressing my concerns and including the discussion on possible scaling directions.

---

### Official Review · Reviewer_C1jB · 2022-07-14

**Rating:** 8
**Confidence:** 4
**Soundness:** 4 excellent
**Presentation:** 4 excellent
**Contribution:** 4 excellent

**Summary:**

Recent studies on controllable text generation often fall short of complex controls like syntactic structure. This paper proposes Diffusion-LM that adopts the idea of denoising diffusion probabilistic models, which have been proven effective, in the vision domain, to naturally incorporate the gradient information from an external classifier for the controls of interest. The main challenge of applying diffusion models to text generation is the discrete nature of language data. The authors propose to introduce an embedding step and its reverse rounding step as the solution. Based on such a remedy, the rest of the model is constructed and trained very similarly to the original DDPM model. Plus, the authors also propose an alternative objective to reduce rounding errors by reparameterizing the model to explicitly predict $x_0$ at every timestep.

Experiments on two datasets and six different control tasks show that the proposed diffusion-LM yields acceptable perplexity and superior performance for controllable generation.

**Questions:**

- Inaccurate terminology. For controls like syntactic structures, the control signal comes from an external parser. I hence suggest using another term rather than “classifier” for sources of control signals.

**Limitations:**

Most limitations of this paper have been discussed in the paper.

**Strengths And Weaknesses:**

Strengths:

1. Very novel methodology contribution that introduces continuous diffusion probabilistic models into text domain with domain-specific adaption. To the best of my knowledge, this is one of the first (or early) successful attempts to study diffusion models for language generation.
2. Impressive performance on controllable text generation, which may motivate further studies on this research problem.
3. The paper is well-organized and mostly clearly described.

Weaknesses:

1. Yet inefficiency in decoding.
2. The model runs upon sequences with short and fixed lengths (64 tokens), which may prevent the approach from generalizing to more practical cases. Are there any results about longer or more flexible/dynamic length choices?

---

> ### Author Response · Authors · 2022-08-02
> **Responses to Reviewer C1jB**
>
> Dear reviewer, thank you for the insightful comments. We are glad you appreciate the novelty and potential impact of Diffusion-LM. We address your comments below.
>
> ----
> [weakness 1]
> We believe the main contribution of this paper is to introduce the new language model class (diffusion-lm) and demonstrate their initial success for controllable generation, and we believe speed-up decoding is best addressed as future work. In this paper, we explored some simple heuristics like down-sampling the diffusion steps at decoding time, which improves decoding time by 10x (although decoding time still remains slower than autoregressive LMs).
> Similarly in the vision case, when DDPM came out, the generation speed was significantly slower than GANs. But some recent work has explored techniques to expedite decoding (Denoising Diffusion Implicit Models https://arxiv.org/abs/2010.02502; Progressive Distillation for Fast Sampling of Diffusion Models https://arxiv.org/abs/2202.00512), and we hope to adopt these advances in vision to NLP in future work.
>
> ----
> [weakness 2]
> Much like autoregressive models, diffusion-lm sets a maximum sequence length and can terminate the sequence at any point.  Currently the model can generate EOS tokens earlier and be able to choose any sequence length smaller than 64. In early preliminary experiments, we experimented with generalizing to length 100 on synthetic data, and the results seem  fluent and achieve similar quality as models trained with length=64.
>
> While extremely long generation poses a challenge compared to autoregressive models using a sliding window,  image diffusions demonstrate that techniques such as generating a coarse, low-resolution image first and applying a super-resolution step to scale the resolution upwards can be an effective way to generate very high dimensional data. We think there are many interesting explorations in this direction and we can investigate the equivalent of super-resolution for text data in future work.
>
> ----
> [Question 1]
> We originally used the term ‘classifier’ for consistency with prior works, which focused only on binary attribute controls. We agree that our syntax experiments involve a parser that is better referred to as a ‘predictor’ than a ‘classifier’ . We will clarify this difference in the main text in the revision.

---

> > ### Comment · Reviewer_C1jB · 2022-08-08
> > **Thank you for responses.**
> >
> > Thank you for your time and effort in addressing my major concerns. I enjoy reading this paper.
> >
> > Best,
> >
> > Reviewer C1jB

---

### Official Review · Reviewer_GmR4 · 2022-07-14

**Rating:** 7
**Confidence:** 5
**Soundness:** 3 good
**Presentation:** 1 poor
**Contribution:** 3 good

**Summary:**

This paper seems to be the first application of diffusion models for (discrete) language data. To deal with the discrete nature, they propose, to my understanding, a softmax layer that maps dense vectors into a distribution over the vocabulary. Another proposed technique is to force the neural network to predict x_0 in every term. The authors test the proposed diffusion models on several datasets and show better results than a set of baseline methods.

**Questions:**

1. Lots of important details are given in Appendix, including the modeling choices for realizing all functions and I really do not know how the classifiers for these control tasks are implemented. I also don't know how infilling is completed.
2. Although this work makes a successful use of diffusion methods. A question is: why would one study diffusion model in the first place? The authors mainly compare diffusion models with autoregressive models to illustrate their advantages. But how about non-autoregressive  models such as https://aclanthology.org/D19-1633.pdf ? I feel non-autoregressive models also (1) enjoy continuous latent representations,  (2) generate all tokens in parallel. and (3) induce a coarse-to-fine hierarchy. In fact, the first and last points are rather vague since any multi-layer neural network can be interpreted in such ways. I feel the discussion with iterative non-autoregressive generation is insufficient.

**Strengths And Weaknesses:**

Strengths:
1. The proposed methods seems effective compared to previous plug-and-play controllable generation approaches.
2. People may be interested in such a successful application of diffusion methods in NLP.
3. The proposed methods are simple and well-examined by ablation studies.

Weaknesses:
1. The report of computational cost is missing (though the authors mentioned that the decoding is substantially slower).
2. I feel it is better to show some human evaluation results regarding the results in Table 1.
3. Might not be a (major) weakness. But I have a hard time reading this paper. The writing is very abstract. I feel important deduction and concrete design are always missing.

---

> ### Author Response · Authors · 2022-08-02
> **Responses to Reviewer GmR4**
>
> Dear reviewer, thank you for the insightful comments. We appreciate your time and effort. We are glad you like our Diffusion-LM and believe people would be interested in the application of diffusion in NLP.  We address your comments below.
>
> ----
> [Weakness 1]  Thanks for the question. We add a compute cost section in the appendix (Appendix C), reporting the compute cost (clock time on a single GPU) to train and decode / control diffusion-lm. We also copy the content in this reply:
>
> Training diffusion LM (we trained it for 200k steps) takes around 8h on a single GPU. Training an autoregressive LM takes around 2h.
>
> Without control, sampling from a Diffusion-LM takes 63 seconds to obtain a batch of 50 samples.  We experiment with some speed up by downsampling diffusion steps from 2000 to 200 at decoding time. The faster version takes 5.6s to obtain a batch of 50 samples. An autoregressive LM takes 0.7s to obtain a batch of 50 samples.
>
> Controllable generation of 50 samples takes around 80s for Diffusion-LM, 4800s for (unbatched) PPLM and 50s for FUDGE.
>
> ----
> [Weakness 2]
> We add human evaluation for the semantic content control experiments to show that our method achieves high fluency and also to justify that our lm-score metrics are quite consistent with human evaluation. This human evaluation result is added to the appendix D (Human Evaluation).
>
> We ask people to compare the outputs of Diffusion-LM and baseline methods, and report which sentence is more fluent. Out of 150 comparisons, we report the percentage of time that users prefer our system over the baseline in the following table. The results suggest that our system is similar or better than the baselines while achieving significantly higher success rate in controls.
>
>
> | A v.s. B                    | A is better | B is better | similar |   |
> |-----------------------------|-------------|-------------|---------|---|
> | Diffusion-LM v.s. FUDGE     | 26.7%       | 8.0%        | 65.3%    |   |
> | Diffusion-LM v.s. FT-sample | 24.0%        | 8.0%         | 68.0%    |   |
>
> ----
> [Weakness 3]
> Due to space constraints, we have to put many details in the appendix. However, with the additional page for publication, we will put some details about the control tasks back to the main text. We have also now added concrete descriptions of control tasks (including the examples of each task specification and how we train the classifier) in the appendix F (Classifier-Guided Controls).
>
> ----
> [Question 1.1]
> How are the classifiers trained?
> To train the classifier for each control task, we first collect sentences and labels.  For example, in a syntactic control task, we use E2E dataset and apply Berkeley parser to get the label parse. The training data is { (sentence, parse) }_i=1^n. The architecture of the parser is a small Transformer model that assigns representation to each token. To get span representation, we concatenate the representation of the left and right boundary tokens.  Finally, we apply a final MLP that takes in each span representation and assigns probabilities over parser labels.
>
> Similarly, for semantic content control tasks, we first collect the (sentence, attribute, value) triples as training data. The classifier structure is based on an autoregressive Transformer model that takes in “<sentence> <eos> <attribute> <eos>” and outputs “<value>”. It’s analogous to an abstractive QA model.
>
> We include the details of POS control and syntactic span controls in the Appendix F.2
>
> [Question 1.2]
> How infilling is done?
> Infilling does not require a classifier, it’s exploiting the flexible factorization of diffusion models to directly sample from the p(infix | prefix, suffix). We follow standard practice in vision (SDEdit: Guided Image Synthesis and Editing with Stochastic Differential Equations https://arxiv.org/abs/2108.01073 ) by using the diffusion backward process to generate the infix, while fixing the prefix and suffix to noisy versions of the input. The pseudo code is as follows:
>
> ```
> x_T(infix) ~ Gaussian(0, I)
> For t in [T-1 … 0]:
>     use the forward process to inject noise to prefix and suffix, yielding x_{t+1}(prefix), x_{t+1}(suffix).
>     use the backward process to denoise the following sequence  [x_{t+1}(prefix), x_{t+1}(infix) x_{t+1}(suffix)] obtaining the denoised sequence [x_t’(prefix), x_t’(infix) x_t’(suffix)]
>     Set x_t(infix) =  x_t’(infix)
> Return x_0(infix)
> ```
> To make the reading clear, we add clear citations to the algorithm in section 5.2 when we discuss infilling experimental setup. We have also added the pseudo code to the appendix G (Infilling).

---

> > ### Author Response · Authors · 2022-08-02
> > **Responses to Reviewer GmR4 (thread 2)**
> >
> > [Question 2]
> >
> > Thank you for bringing this up. It is certainly true that some NAR models also satisfy some of these three properties (continuous, non-autoregressive, coarse-to-fine).  In our revision, we have added more careful definitions of these terms and an extended discussion of NAR models and their role in controllable generation in the appendix, and will reference this discussion in the main text.
> >
> > With regard to NAR models in particular, we did not extensively study or compare to these models in our work, as all of the existing high-performance controllable generation methods are autoregressive. NAR methods such as Mask-Predict are non-autoregressive and sometimes hierarchical, but often do not admit a fully continuous representation on which we can apply the gradient based control methods used in our work (in particular, Mask-Predict can be thought of as repeatedly rounding to discrete tokens).
> >
> > We believe that comparing to NAR models (and discrete diffusions) in controllable generation settings would be interesting, but it is something we did not do for this work, as it would involve the development of entirely new plug-and-play control methods that can operate on discrete, NAR models such as Mask-Predict.

---

### Author Response · Authors · 2022-08-02
**General Reply to All Reviewers**

We thank all the reviewers for their detailed feedback.  We have incorporated suggestions into the appendix, which will appear in the next version of the main paper.

We are glad the reviewers appreciate the novelty and impact of our work. The reviewers recognize our technical contribution as the first work to apply continuous diffusion models to text data. The reviewers also appreciate the empirical results demonstrating effectiveness of Diffusion-LM for controllable generation tasks.

Reviewers point out that Diffusion-LM shares the general drawbacks of diffusion models: slow decoding and training speed compared to autoregressive LMs.
While this is true, we believe that enabling diffusion models to operate on text data (Diffusion-LM), and demonstrating substantial gains on a range of controllable generation tasks addresses the major modality-specific challenges, and we believe that speeding up decoding/training is interesting future work to be done jointly with the image diffusion community.

In the case of image diffusions, when DDPM came out, the generation speed was significantly slower than GANs. But many follow-up works have explored techniques to expedite decoding (e.g., Denoising Diffusion Implicit Models https://arxiv.org/abs/2010.02502). Work has also been done in the vision domain to scale up the model sizes (e.g., DallE-2 https://arxiv.org/abs/2204.06125; Imagen https://arxiv.org/abs/2205.11487 ). Given these recent successes and interest in scaling image diffusions, as well as improvements to decoding speed, we are optimistic that further work along these lines will improve Diffusion-LM’s computational drawbacks over time.

---

### Meta-Review · Area_Chair_XHbF · 2022-08-20

**Recommendation:** Accept
**Confidence:** Certain

**Metareview:**

This paper introduces a new non-autoregressive language model based on continuous diffusion models, which have been quite successful in the vision area. All reviewers agree that the method is novel, this paper is well-written, and the experiments are convincing. Although there are concerns about the efficiency, this paper is worthy of being accepted by NeuRIPS.

**Award:**

No

---

### Decision · Program_Chairs · 2022-09-14

Accept